# NEURAL PROGRAM SYNTHESIS WITH QUERY

**Di Huang**[1,2,4], **Rui Zhang**[1,4]*, **Xing Hu**[1,4], **Xishan Zhang**[1,4], **Pengwei Jin**[1,2,4],
**Nan Li**[1,3,4], **Zidong Du**[1,4], **Qi Guo**[1] **& Yunji Chen**[1,2]
[1]SKL of Computer Architecture, Institute of Computing Technology, CAS
[2]University of Chinese Academy of Sciences
[3]University of Science and Technology of China
[4]Cambricon Technologies

## ABSTRACT

Aiming to find a program satisfying the user intent given input-output examples, program synthesis has attracted increasing interest in the area of machine learning. Despite the promising performance of existing methods, most of their success comes from the privileged information of well-designed input-output examples. However, providing such input-output examples is unrealistic because it requires the users to have the ability to describe the underlying program with a few input-output examples under the training distribution. In this work, we propose a query-based framework that trains a query neural network to generate informative input-output examples automatically and interactively from a large query space. The quality of the query depends on the amount of the mutual information between the query and the corresponding program, which can guide the optimization of the query framework. To estimate the mutual information more accurately, we introduce the *functional space (F-space)* which models the relevance between the input-output examples and the programs in a differentiable way. We evaluate the effectiveness and generalization of the proposed query-based framework on the Karel task and the list processing task. Experimental results show that the query-based framework can generate informative input-output examples which achieve and even outperform well-designed input-output examples.

## 1 INTRODUCTION

Program synthesis is the task of automatically finding a program that satisfies the user intent expressed in the form of some specifications like input-output examples (Gulwani et al., 2017). Recently, there has been an increasing interest in tackling it using neural networks in various domains, including string manipulation (Gulwani, 2011; Gulwani et al., 2012; Devlin et al., 2017b), list processing (Balog et al., 2017; Zohar & Wolf, 2018) and graphic applications (Ellis et al., 2018; 2019).

Despite their promising performance, most of their success relies on the well-designed input-output examples, without which the performance of the program synthesis model drops heavily. For example, in Karel (Devlin et al., 2017b; Bunel et al., 2018), the given input-output examples are required to have a high branch coverage ratio on the test program; In list processing, the given input-output examples are guided by constraint propagation (Balog et al., 2017) to ensure their effectiveness on the test program. However, providing such input-output examples is unrealistic because it requires the users to be experienced in programming to describe the underlying program with several input-output examples. Worse, the users must be familiar with the distribution of the training dataset to prevent themselves from providing out-of-distribution examples (Shin et al., 2019). In summary, how to generate informative input-output examples without expert experience is still an important problem that remains a challenge.

In this paper, we propose a query-based framework to automatically and efficiently generate informative input-output examples from a large space, which means that the underlying program can be easily distinguished with these examples. This framework consists of two parts: the query network

---

*Corresponding author. Contact: {huangdi20b, zhangrui}@ict.ac.cn.

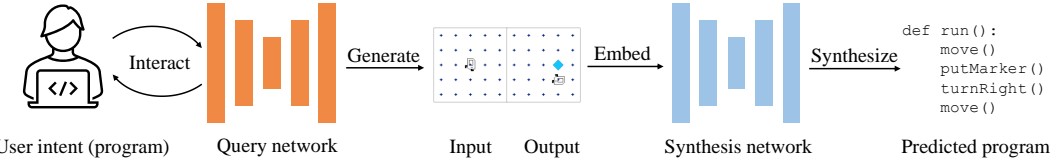

Figure 1: The query-based framework. The query network generates informative input-output examples by interacting with the user, and then the generated examples are sent to the synthesis network to synthesize the underlying program.

and the synthesis network, and each part is trained separately. The query network is trained to generate the input-output examples in an iterative manner, and then the synthesis network is trained to synthesize the program with the input-output examples generated by the query network. It has three advantages: (1) There is no demand for well-designed input-output examples in both training and testing, which leads to a good generalization. (2) The query network works in an efficiently generative manner, which can essentially reduce the computation cost while facing problems with large input-output space. (3) The query network serves as a plug-and-play module with high scalability, for it is separated from the program synthesis process entirely.

To train the query network, the key idea is to model the query process as a decision tree and find out that the informativeness of the input-output examples is associated with the amount of mutual information (*i.e.*the information gain in the decision tree) between the input-output examples and the programs, and thus the query network can be optimized by maximizing the mutual information. The mutual information can be approximated by the InfoNCE loss (van den Oord et al., 2018), which depends on the relevance between the samples. However, due to the many-to-many relationship between the input-output examples and the programs, the relevance is difficult to be measured straightforwardly. To this end, we introduce the *functional space (F-space)*. In F-space, each program can be projected into a vector, and each input-output example corresponds to a set of candidate programs, which can be represented by a normal distribution (Sun & Nielsen, 2019). Whenever a new example is queried, the new set of candidate programs is produced by the intersection of the two original sets of programs, and the new distribution is produced by the product of the two original distributions. The more the queried examples, the smaller the size of the program set, the lower the entropy of the distribution. With the InfoNCE loss and F-space, the query network is trained to generate the input-output examples whose F-space distribution can maximize the probability of the corresponding program and minimize the probability of the others. Once the query network is trained, the training of the synthesis network is no different from what other researchers have done before, except that the input-output examples are generated by the query network.

We evaluate our method on the Karel task and the list processing task which have large input spaces. Without utilizing the well-designed input-output examples both in training and testing, we achieve and even outperform the state-of-the-art performance on the Karel task and list processing task, which shows the effectiveness and generalization of our query-based framework.

## 2 PROBLEM STATEMENT

Intuitively, consider an oracle that contains some kind of symbolic rules (*e.g.*programs), our goal is to discover these symbolic rules inside this oracle. To do this, the traditional program synthesis assumes that the oracle can provide some informative signals (*e.g.*input-output examples) automatically, based on which the synthesizer can find the rules. However, this is a strong assumption with high requirements on the oracle that is impractical in many cases. In this work, we consider a query problem where the informative signals are gained actively by querying the oracle under a much weaker assumption that the behavior of the oracle is deterministic (*i.e.*the same input always results in the same output).

Following the intuition, a reasonable solution for the query problem would be "making the programs distinguishable with as few queries as possible". To make this statement concrete and practical, we introduce the following formulations.

First, we define the functional equivalence, which is consistent with the definition of the equivalence of functions in mathematics.

**Definition 2.1** (Functional equivalence). *Let $\mathbb{I}$ be the input example domain containing all valid input examples, $\mathbb{O}$ be the output example domain containing all possible output examples, and $\mathbb{P}$ be the program domain containing all valid programs under the domain specific language (DSL), each program $p \in \mathbb{P}$ can be seen as a function $p : \mathbb{I} \to \mathbb{O}$. For two programs $p_i \in \mathbb{P}$ and $p_j \in \mathbb{P}$, $p_i$ and $p_j$ are functional equivalent if and only if $\forall x \in \mathbb{I}, p_i(x) = p_j(x)$.*

Using the concept of functional equivalence, we can formulate the program synthesis task:

**Definition 2.2** (Program synthesis). *Suppose there is an underlying program $p \in \mathbb{P}$ and $K$ input-output examples $[\![e]\!] = \{(x_k, y_k)|(x_k, y_k) \in \mathbb{I} \times \mathbb{O}, k = 1 \cdots K\}$ generated by it, the program synthesis task aims to find a program $\hat{p}$ which is functional equivalent to $p$ using $[\![e]\!]$.*

Following the definitions above, we can define the task of the query.

**Definition 2.3** (Query). *Given a program $p \in \mathbb{P}$ and a set of history $K$ input-output examples $[\![e]\!]$, the query process firstly generates a query by the query network $f_q$: $x_{K+1} = f_q([\![e]\!]) \in \mathbb{I}$. Then, a corresponding response is given by the program $y_{K+1} = p(x_{K+1}) \in \mathbb{O}$. The query and response are added to the history input-output examples: $[\![e]\!] \leftarrow [\![e]\!] \cup \{(x_{K+1}, y_{K+1})\}$ and this process repeats.*

The query process aims to distinguish the underlying program with as few queries as possible, based on which we can define the optimal query strategy.

**Definition 2.4** (The optimal query strategy). *Given a set of programs $\mathbb{P}$ and a target program $p^*$, the optimal query strategy $Q$ aims to distinguish the $p^*$ by generating as few input-output examples $[\![e]\!]$ as possible.*

When the query space is large, it is impractical to find the optimal query strategy by direct search. Thus, we take the query process as the expansion of a decision tree where the leaves are the programs, the nodes are the queries, and the edges are the responses, and then the generation of queries can be guided by the information gain, which is also known as the mutual information:

$$[\![e]\!]^* = \arg\max_{[\![e]\!]} I(\mathbb{P}; [\![e]\!]), \tag{1}$$

## 3 METHODS

The mutual information is hard to calculate due to the large spaces of queries and programs. To this end, we resort to the InfoNCE loss (van den Oord et al., 2018) which estimates the lower bound of the mutual information:

$$L_{NCE} = -\mathbb{E}[log(\frac{exp(f([\![e]\!], p_n))}{\sum_{i=1}^{N} exp(f([\![e]\!], p_i))})], \tag{2}$$

and

$$I(\mathbb{P}; [\![e]\!]) \geq log(N) - L_{NCE}, \tag{3}$$

where $f(\cdot, \cdot)$ denotes a relevance function. Maximizing the mutual information is equivalent to minimizing $L_{NCE}$. Intuitively, InfoNCE maximizes the relevance between the positive pairs and minimizes the relevance between the negative pairs. Specifically, for a batch of data $\{([\![e]\!]_n, p_n), p_n\}^N$, we construct positive pairs as $\{([\![e]\!]_i, p_i)\}$ and negative pairs as $\{([\![e]\!]_i, p_j)|i \neq j\}$. Traditionally, the relevance is defined as the dot product of the samples, which may be inaccurate for the many-to-many relationship between the input-output examples and the programs. Thus next, we will introduce *functional space (F-space)* to model the relationship properly.

### 3.1 F-SPACE

**Definition 3.1** (F-space and functional distance). *F-space is a $|\mathbb{I}|$ dimensional space which consists of all valid programs that can be implemented by program domain $\mathbb{P}$. Each program is represented by $|\mathbb{I}|$ different output examples $\boldsymbol{v} = (y_1, y_2, \ldots, y_{|\mathbb{I}|})$. The distance in F-space can be measured by the number of different output examples: $d(\boldsymbol{v}_i, \boldsymbol{v}_j) = |diff(\boldsymbol{v}_i, \boldsymbol{v}_j)|$.*

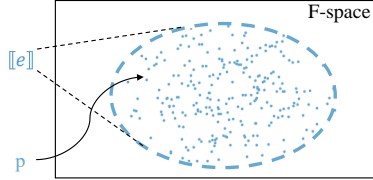 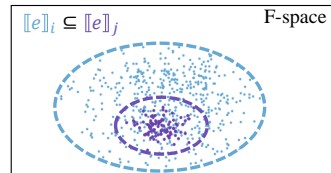 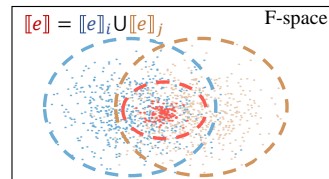

Figure 2: The illustration of F-space. Left: The projection of the input-output examples $[\![e]\!]$ and program $p$; Middle: The subset relationship between $[\![e]\!]_i$ and $[\![e]\!]_j$. Note that this relation is opposite in F-space; Right: The union operation of $[\![e]\!]_i$ and $[\![e]\!]_j$. This operation is also opposite in F-space where an union operation correspond to an intersection operation in F-space.

Intuitively, F-space $(\mathbb{P}, d)$ measures the functional differences between programs. Each program can be represented by a vector in F-space, and if different programs are represented by the same vector $\boldsymbol{v} = (y_1, y_2, \ldots, y_{|\mathbb{I}|})$, it indicates that these two programs get the same results for all inputs, and they are considered to be functionally equivalent. This is also consistent with Definition 2.1. In practice, a space with dimension $|\mathbb{I}|$ is too large to compute, and thus we utilize the sparsity of F-space and learn an approximate space with dimension reduction by neural networks.

Regarding input-output examples, representing them by vectors is not appropriate. Consider a set of input-output examples $[\![e]\!] = \{(x_k, y_k)\}^K$, we cannot find a vector that represents it when $K < |\mathbb{I}|$ because there are more than one programs that satisfies these $K$ input-output examples.

Formally, we summarize the properties of the representation of input-output examples in F-space as follows:

- Each set of input-output examples $[\![e]\!] = \{(x_k, y_k)\}^K$ should be represented by a set of F-space vectors $[\![r]\!] = \{\boldsymbol{v}_n\}^N$.
- For two sets of input-output examples $[\![e]\!]_i$ and $[\![e]\!]_j$, if $[\![e]\!]_i \subseteq [\![e]\!]_j$, then their F-space representations $[\![r]\!]_i$ and $[\![r]\!]_j$ have the relationship $[\![r]\!]_i \supseteq [\![r]\!]_j$.
- For two sets of input-output examples $[\![e]\!]_i$ and $[\![e]\!]_j$, suppose $[\![e]\!]' = [\![e]\!]_i \cap [\![e]\!]_j$, then in F-space, their corresponding representations have the relationship $[\![r]\!]' = [\![r]\!]_i \cup [\![r]\!]_j$
- Similarly, if $[\![e]\!]' = [\![e]\!]_i \cup [\![e]\!]_j$, then in F-space, their corresponding representations have the relationship $[\![r]\!]' = [\![r]\!]_i \cap [\![r]\!]_j$

Additionally, this representation should be differentiable and thus can be optimized by neural networks. To this end, we model the representation of $[\![e]\!]$ as a Normal distribution (Ren & Leskovec, 2020; Sun & Nielsen, 2019), where the probability of the distribution indicates the possibility that it is the underlying program to be synthesized given input-output examples $[\![e]\!]$. We illustrate the projection, the subset relationship, and the union/intersection operation of distributions in Figure 2. Under this representation, the query process becomes the process of reducing the uncertainty of the distribution. To train the query network, we define several neural operators for the neural network training as follows.

**Program projection.** To project a program $p$ to a vector $\boldsymbol{v}$ in F-space, we traditionally use a sequence model as our encoder:

$$\boldsymbol{v} = Encoder_p(p). \tag{4}$$

Note that the program synthesis model differs largely on different datasets, so we choose to vary our program encoder according to the specific program synthesis models on different datasets. In practice, our program encoder is the reverse of the program synthesis decoder.

**Input-output example projection.** Given a single input-output example $[\![e]\!] = \{(x, y)\}$, we can project it into a Normal distribution using the same architecture of the corresponding program synthesis model, except that we add an MLP to output the two parameters of the Normal distribution: $\boldsymbol{\mu}$ and $log(\boldsymbol{\sigma}^2)$.

$$[\boldsymbol{\mu}, log(\boldsymbol{\sigma}^2)] = MLP_e(Encoder_e([\![e]\!])), \tag{5}$$

where $MLP$ means a multi-layer perceptron.

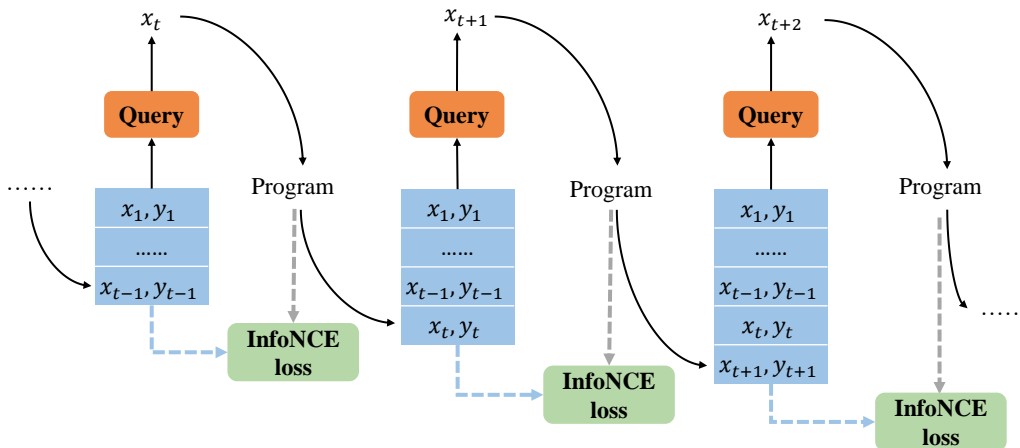

Figure 3: The recurrent training process.

**Input-output examples intersection.** Given $K$ input-output examples $[\![e]\!] = \{(x_k, y_k)\}^K$, each example $\{(x_k, y_k)\}$ is represented by a Normal distribution $Pr_k = \mathcal{N}(\boldsymbol{\mu}_k, \boldsymbol{\sigma}_k)$ using the projector above. The purpose of the intersection is to aggregate these Normal distributions into a new one, which represents $[\![e]\!]$ in F-space. Under the assumption of independence, the probability of the intersection distribution should be:

$$Pr_{[\![e]\!]} = \prod_{k=1}^{K} Pr_k. \tag{6}$$

Fortunately, the product of independent Normal distributions is still a Normal distribution, which means that we can represent it as $[\boldsymbol{\mu}', log(\boldsymbol{\sigma}'^2)]$. In practice, we utilize the attention mechanism with another MLP to let the neural network learn the new Normal distribution (Ren & Leskovec, 2020):

$$[\boldsymbol{\mu}', log(\boldsymbol{\sigma}'^2)] = \sum_{i=1}^{K} w_i[\boldsymbol{\mu}_i, log(\boldsymbol{\sigma}_i^2)], \tag{7}$$

$$w_i = \frac{exp(MLP_{attention}([\boldsymbol{\mu}_i, log(\boldsymbol{\sigma}_i^2)]))}{\sum_{j=1}^{K} exp(MLP_{attention}(\boldsymbol{\mu}_j, log(\boldsymbol{\sigma}_j^2)))}. \tag{8}$$

This formulation not only keeps the form of distributions closed but also approximates the mode of the effective support of Normal distributions, which satisfies our requirement on the intersection of distribution (Sun & Nielsen, 2019). We give detailed proof on the reasonability of doing this in Appendix C.1.

**Inverse projection to query.** Finally, we need to generate a new query from the representation of input-output examples in F-space. Using the projection and the intersection operation mentioned above, we can project the K input-output examples into the representation $[\boldsymbol{\mu}', log(\boldsymbol{\sigma}'^2)]$. To generate query $x_{new}$ from this representation, we introduce a decoder which has a similar architecture to $Encoder_e$ but reversed:

$$x_{new} = Decoder([\boldsymbol{\mu}', log(\boldsymbol{\sigma}'^2)]). \tag{9}$$

With these operators, we can model the relevance between input-output examples and programs as the probability in the distributions in F-space, and rewrite the loss in Equation 2 as

$$L_{NCE} = -\mathbb{E}[log(\frac{exp(\mathcal{N}(p_n; \boldsymbol{\mu}', \boldsymbol{\sigma}'))}{\sum_{i=1}^{N} exp(\mathcal{N}(p_i; \boldsymbol{\mu}', \boldsymbol{\sigma}'))})], \tag{10}$$

Next, we will introduce how to train the neural network.

## 3.2 TRAINING

As mentioned above, the query network is optimized by maximizing the mutual information. We observe that taking previous query examples as the condition and only optimizing the query strategy at the current query step is a greedy optimization, which may fail into the local minima more easily. An example is shown in Appendix C.2. Thus, a recurrent training process is adopted to grep the global mutual information on every step instead of the mutual information conditioned on the previous examples. See Figure 3 and Algorithm 1.

Additionally, like the task of sequence generation, we need an input-output example to act as the start signal $< sos >$. However, different datasets differ largely in program synthesis, which makes the design of a universal start signal difficult. Thus, we design specific start signals for each dataset separately. For example, in Karel, the signal is designed to be an empty map with the robot at the center of the map; In list processing, the signal is just three lists full of *NULL*. More training details such as the network architectures, the processing of datasets, training tricks, can be seen in Appendix A.

## 4 EXPERIMENTS

We studied three problems in our experiments. (1) The metric that is reasonable in our query-based framework. (2) The performance of the query-based framework. (3) The generalization ability on different tasks of the query-based framework. To do this, first, we demonstrate a comparison among several program synthesis metrics. Then, we present our results of the query on the Karel dataset and the list processing dataset to show our methods' performance and generalization ability.

### 4.1 METRICS

There are three metrics in neural program synthesis.

- **Semantics**: Given 5 input-output examples, if the predicted program satisfies all these examples, then it is semantically correct.
- **Generalization**: Given 5 input-output examples and a held-out input-output example, if the predicted program satisfies all 6 examples, then it is generally correct.
- **Exact match**: If the predicted program is the same as the ground-truth program, then it matches the ground-truth exactly.

Among them, the exact match is the most strict metric and generalization is the second one. In practice, the exact match has its drawback in that the predicted program may be different from the ground-truth program, but they are functionally equivalent. Thus, the performance is measured by generalization on Karel and semantics on list processing traditionally. However, generalization is not appropriate here for two reasons: (1) It can only measure the performance of the program synthesis process instead of the query process. In the query process, the network chooses input-output examples independently, and judging them by the held-out example is meaningless. (2) Even for the program synthesis process without query, generalization is not strict enough. The program that satisfies a small set of input-output examples may fail on a larger set of input-output examples. Thus, the best choice is to use *functional equivalence* as our metric. Unfortunately, the judgment of functional equivalent is a notoriously difficult problem which makes it hard to be applied in evaluation. To alleviate this problem, we generate 95 held-out input-output examples randomly to achieve a higher branch coverage than 1 held-out example, and make the generalization on these examples a proxy of the functional equivalence:

- **Functional equivalence (proxy)**: Given 5 input-output examples and 95 held-out input-output examples, if the predicted program satisfies all 100 examples, then it is functional equivalent to the ground-truth program.

To illustrate the difference between functional equivalence and generalization further, we measured the average branch coverage of these two metrics on Karel's validation set. Given a set of input-output examples, branch coverage evaluates the percentage of program branches that are covered by the examples.

Table 2: The performance of the program synthesis model trained on input-output examples generated by different methods on the Karel task.

| Metric | Bunel et al. (2018) | | | Chen et al. (2019) | | |
|---|---|---|---|---|---|---|
| | Random | Well-designed | Query | Random | Well-designed | Query |
| Exact match | 29.44% | 41.16% | 41.12% | 26.08% | 37.36% | 38.68% |
| Functional equivalence | 33.08% | 48.52% | 46.64% | 32.28% | 47.60% | 48.48% |

The result is shown in Table 1. Functional equivalence outperforms generalization by nearly 10%, which indicates that functional equivalence can represent the correctness of the predicted program much better than generalization.

| | Semantics | Generalization | FE |
|---|---|---|---|
| Branch coverage | 86.57% | 87.99% | 97.58% |

Table 1: The branch coverage of semantics, generalization and functional equivalence (FE).

## 4.2 KAREL TASK

Karel is an educational programming language used to control a robot living in a 2D grid world (Pattis et al., 1981). The domain-specific language (DSL) and other details are included in Appendix B.1.

**Settings.** Following Section 3, we split the training process into the training of the query network and the training of the synthesis network. For the query network, we set $Encoder_e$ the same as the one of Bunel et al. (2018) except for the output layer, and $Encoder_p$ a two-layer LSTM (see details in Appendix A.2). For the synthesis network, all settings stay unchanged as in the original program synthesis method. We report the top-1 results with beam size 100 for Bunel et al. (2018) and 64 for Chen et al. (2019) which is the default setting for validation in the original code. As in Section 4.1, exact match and functional equivalence are more appropriate than other metrics in the query task, so we save the checkpoints based on the best exact match instead of the best generalization (for the computation inefficiency of functional equivalence), which may cause differences in our baseline reproduction.

We generate the dataset with 5 input-output examples using three different methods: randomly selected (random), baseline models (well-designed), and our method (query). Then, we train the synthesis network on these datasets with two state-of-the-art methods: Bunel et al. (2018) and Chen et al. (2019). Note that there is a slight difference between the program simulators used by them which will give different responses during the query process, and we choose the one in Bunel et al. (2018) as ours.

**Dataset Performance.** Table 2 presents the performance of the trained synthesis networks, from which we can conclude that (1) The queried dataset performs well on both two training methods, indicating that the query process is totally decoupled with the synthesis process and has a high generality. (2) Our query method gets comparable results with both baseline methods and largely outperforms the random selection method with more than 10%, which shows its effectiveness.

**Comparison on query.** We also compared our method with query by committee (QBC) Seung et al. (1992) as another baseline, shown in Figure 4. In QBC, we sample queries based on their diversity. That is, we generate program candidates by beam search, and then select the query that can result in the most diverse outputs on these program candidates. The diversity is measured by the output equivalence. Algorithm details can be seen

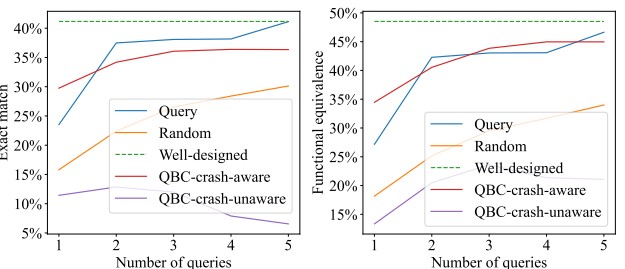

Figure 4: The query performance of different methods.

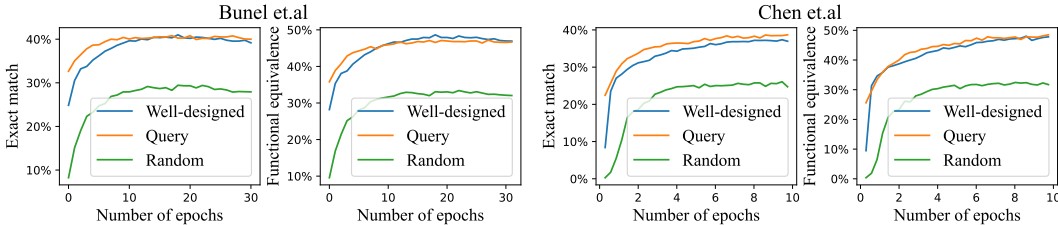

Figure 5: The training curve of the program synthesis model on Karel.

Table 3: The performance of the program synthesis model trained on input-output examples generated by different methods on the list processing task.

| Dataset | Metric | Searching for semantics | | | Searching for exact match | | |
|---------|--------|--------|---------------|-------|--------|---------------|-------|
| | | Random | Well-designed | Query | Random | Well-designed | Query |
| $D_1$ | Exact match | 20.07% | 32.81% | 22.26% | 50.65% | 81.21% | 81.56% |
| | Functional equivalence | 52.03% | 80.26% | 68.20% | 50.65% | 81.21% | 81.56% |
| $D_2$ | Exact match | 9.25% | 15.72% | 10.44% | 22.61% | 38.51% | 38.63% |
| | Functional equivalence | 24.49% | 39.88% | 32.94% | 22.61% | 38.51% | 38.63% |

in Appendix B.4. There are two strategies based on QBC: Crash-aware, which repeats the algorithm to sample another one if the query crashes; And crash-unaware, which samples queries regardless of the crash problem (see Appendix B.1 for a detailed explanation of crashes). QBC-crash-unaware performs worse than Random because Random is chosen by filtering crashed IOs while QBC-crash-unaware may contain crashes. QBC-crash-aware performs much better than QBC-crash-unaware because it queries the underlying program multiple times to make sure that the query will not result in a crash, which is unfair. Even though, out method still outperforms QBC-crash-aware, which shows its advantage.

**Training process.**  To do a further study, we plot the training curve in Figure 5. It is shown that the Query dataset always has a quick start, which indicates that the query method can extract the features of the distinguishable programs effectively and makes them easier to be synthesized.

### 4.3  LIST PROCESSING TASK

To show the generalization ability of the query method, we conduct another experiment on the list processing task. The list processing task takes 1-3 lists or integers as the input example, and then produces a list or an integer as the output example. More details can be seen in Appendix B.2.

**Settings.**  Following PCCoder (Zohar & Wolf, 2018), we generate two datasets with program length 4 as dataset $D_1$ and program length up to 12 as dataset $D_2$. We set the $Encoder_e$ of the query network similar to PCCoder except for an MLP, and use a single layer LSTM as the $Encoder_p$ (see Appendix A.2). The synthesis network and the parameters of complete anytime beam search (CAB) (Zhang, 1998) stay the same as PCCoder, except that the maximum time is set to 5 seconds instead of 5,000 seconds for reality.

Similar to the Karel task, we generate the dataset with 5 input-output examples with three different methods. Furthermore, we use two end conditions of CAB: The searching ends when all input-output examples are satisfied (searching for semantics), and the searching ends when all statements are true (searching for exact match).

**Performance.**  The results are presented in Table 3, from which we can conclude that: (1) Our query method results higher than well-designed input-output examples in searching for the exact match and consistently outperforms the random. (2) When the length of programs increases, the performance decreases largely. This results from the original algorithm of PCCoder that the inter-

mediate variables are difficult to be saved when the ground-truth program is long. However, the query method decreases slower than others, and the gap between the well-designed and the query is closed largely.

## 5 RELATED WORK

**Programming by examples.** Synthesizing a program that satisfies the user intent using the provided input-output examples is a challenging problem that has been studied for years (Manna & Waldinger, 1971; Lieberman, 2001; Solar-Lezama et al., 2006; Gulwani, 2011; Gulwani et al., 2012). Recently, with the development of Deep Learning, more researchers tend to tackle this problem with neural networks in a variety of tasks, including string transformation (Devlin et al., 2017b), list processing (Balog et al., 2017; Zohar & Wolf, 2018), graphic generation (Ellis et al., 2018; Tian et al., 2019), Karel (Devlin et al., 2017a; Bunel et al., 2018), policy abstraction (Sun et al., 2018; Verma et al., 2018) and so on. Additionally, techniques like program debugger (Balog et al., 2020; Gupta et al., 2020), traces (Chen et al., 2019; Shin et al., 2018; Ellis et al., 2019), property signatures (Odena & Sutton, 2020; Odena et al., 2021) are also used to improve the performance of program synthesis. However, their promising performance relies on the well-designed input-output examples which is a high requirement for users. If the examples provided are of poor quality, the performance will be affected severely. Worse, if out-of-distribution examples are provided, the program synthesis model cannot finish the task as expected (Shin et al., 2019).

**Interactive program synthesis.** Considering the high requirement of input-output examples, Le et al. (2017) build an abstract framework in which the program synthesis system can interact with users to guide the process of synthesis. Following this framework, multiple interactive synthesis systems are studied. Among them, Mayer et al. (2015), Wang et al. (2017), and Laich et al. (2020) tend to find counter-examples as queries to the users in a random manner. Although they get rid of the restrictions on the users, the quantity of the input-output examples is not guaranteed, which may get the synthesis process into trouble. Padhi et al. (2018) select more than one query each time and let the users choose which one to answer. This brings an additional burden on the users, and the users are unaware of which query will improve the program synthesized most. Most recently, Ji et al. (2020) utilizes the minimax branch to select the question where the worst answer gives the best reduction of the program domain. Theoretically, the performance of the selection is guaranteed. However, this method is based on search and hardly be applied to tasks with large input-output space. Worse, the query process and the synthesis process are bound, which results in its poor scalability. In contrast, our method can be applied to problems with large spaces, and the query process is decoupled with the synthesis process, which makes its application more flexible.

**Learning to acquire information.** Similar to our work, Pu et al. (2018) and Pu et al. (2017) also study the query problem from an information-theoretic perspective. They show that maximizing the mutual information between the input-output examples and the corresponding program greedily is $1 - \frac{1}{e}$ as good as the optimal solution that considers all examples globally. However, they assume that the space of queries can be enumerated, which limits the application of their query algorithm on complex datasets like Karel. By comparison, our work proposes a more general algorithm that can generate queries in a nearly infinite space. Other related work including active learning and black-box testing can be seen in Appendix D.

## 6 CONCLUSION

In this work, we propose a query-based framework to finish the program synthesis task more realistically. To optimize this framework, we show the correlation between the query strategy and the mutual information. Moreover, we model the relevance between the input-output examples and programs by introducing the F-space, where we represent the input-output examples as the distribution of the programs. Using these techniques, we conduct a series of experiments that shows the effectiveness, generalization, and scalability of our query-based framework. We believe that our methods work not only on the program synthesis tasks, but also on any task that aims to simulate an underlying oracle, including reverse engineering, symbolic regression, scientific discovery, and so on.

ACKNOWLEDGMENTS

This work is partially supported by the National Key Research and Development Program of China (under Grant 2020AAA0103802), the NSF of China (under Grants 61925208, 62102399, 62002338, 61906179, 61732020, U19B2019), Strategic Priority Research Program of Chinese Academy of Science (XDB32050200), Beijing Academy of Artificial Intelligence (BAAI) and Beijing Nova Program of Science and Technology (Z191100001119093), CAS Project for Young Scientists in Basic Research (YSBR-029), Youth Innovation Promotion Association CAS and Xplore Prize.

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

# A  TRAINING DETAILS

## A.1  TRAINING ALGORITHM

---

**Algorithm 1** Training process

---

1: **function** TRAIN( )
2:   Initialize max iterations $N$ and max query times $T$
3:   **for** $i \in \{1 \dots N\}$ **do**
4:     $p \leftarrow Underlying\ programs$
5:     $(x_0, y_0) \leftarrow < sos >$
6:     $[\![e]\!] \leftarrow \{(x_0, y_0)\}$
7:     $L \leftarrow 0$
8:     **for** $t \in \{1 \dots T\}$ **do**
9:       $\boldsymbol{v} \leftarrow Encoder_p(p)$                    ▷ Encode program, Equation (4)
10:       $\boldsymbol{\mu}, log(\boldsymbol{\sigma}^2) \leftarrow$ IO-ENCODER($[\![e]\!]$)       ▷ Encode input-output examples
11:       $x_t \leftarrow Decoder(\boldsymbol{\mu}, log(\boldsymbol{\sigma}^2))$                 ▷ Query next input
12:       $y_t \leftarrow p(x_t)$                   ▷ Get next output from the oracle
13:       $[\![e]\!] \leftarrow [\![e]\!] \cup \{(x_t, y_t)\}$
14:       $\boldsymbol{\mu}', log(\boldsymbol{\sigma}'^2) \leftarrow$ IO-ENCODER($[\![e]\!]$)
15:       $L \leftarrow Loss(\boldsymbol{v}, \boldsymbol{\mu}', log(\boldsymbol{\sigma}'^2)) + L$           ▷ InfoNCE loss, Equation (2)
16:     **end for**
17:     Update parameters w.r.t. $L$
18:   **end for**
19: **end function**

1: **function** IO-ENCODER($\{x_k, y_k\}^K$)
2:   **for** $i \in \{1 \dots K\}$ **do**
3:     $[\boldsymbol{\mu}_i, log(\boldsymbol{\sigma}_i^2)] \leftarrow MLP_e(Encoder_e(\{x_i, y_i\})$   ▷ Encode single example, Equation (5)
4:   **end for**
5:   **for** $i \in \{1 \dots K\}$ **do**
6:     $w_i \leftarrow \frac{exp(MLP_{attention}([\boldsymbol{\mu}_i, log(\boldsymbol{\sigma}_i^2)]))}{\sum_{j=1}^{K} exp(MLP_{attention}(\boldsymbol{\mu}_j, log(\boldsymbol{\sigma}_j^2)))}$   ▷ Calculate attentions, Equation (8)
7:   **end for**
8:   $[\boldsymbol{\mu}, log(\boldsymbol{\sigma}^2)] = \sum_{i=1}^{K} w_i[\boldsymbol{\mu}_i, log(\boldsymbol{\sigma}_i^2)]$            ▷ Intersection, Equation (7)
9:   **return** $\boldsymbol{\mu}, log(\boldsymbol{\sigma}^2)$
10: **end function**

---

## A.2  MODEL DETAILS AND HYPER PARAMETERS

**Karel.**  The query encoder is the same as the one used by Bunel et al. (2018) composed of a Convolutional Neural Network (CNN) with the residual link. After the query encoder, there is an MLP to project the embedding into $\boldsymbol{\mu}, log(\boldsymbol{\sigma}^2)$. The dimension of $\boldsymbol{\mu}$ and $\boldsymbol{\sigma}$ is set to 256, and thus the hidden size of MLP is 512. The query decoder is similar to the query encoder except that the number of channels is reversed, and additional batch normalization is added to keep the generation more stable. The program encoder is a two-layer LSTM with a hidden size of 256.

Note that in the well-designed dataset, the size of the grid world can be changed from 2 to 16. However, for the convenience of training, we set the size of the query world fixed to 16. Moreover, to guarantee that all queries can be recognized by the program simulator, we split the query into three parts: boundaries (the map size, set to 16×16), agent position (where the agent is and towards), and map state (the placement of markers and obstacles), and generate them as follows:

- **Boundaries**: The boundaries indicate the map size, fixed to $16 \times 16$.
- **Agent position**: generate a $4 \times 16 \times 16$ one-hot vector where 4 indicates four facing directions and $16 \times 16$ indicates the position.
- **Map state**: generate a 12 one-hot vector on the $16 \times 16$ map including obstacle, 1-10 markers, and empty grid.

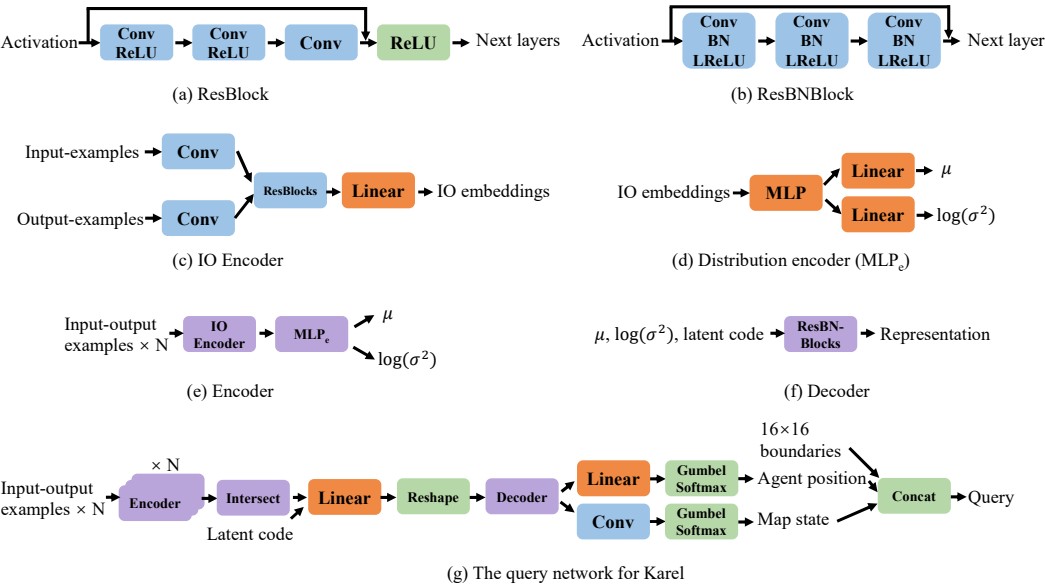

Figure 6: The architecture of the query network for Karel (zoom in for a better view).

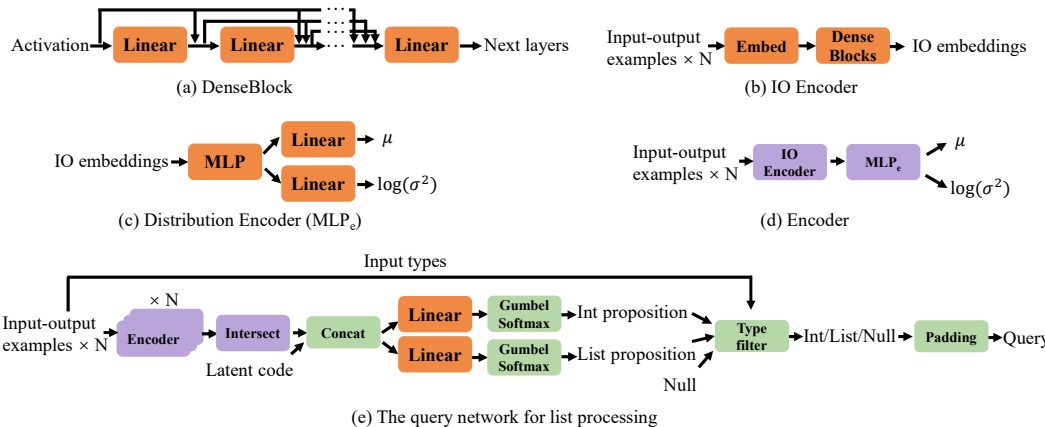

Figure 7: The architecture of the query network for list processing (zoom in for a better view). Type filter chooses query between the list proposition and int proposition depends on the input types.

The architecture details are shown in figure 6.

For training, the learning rate of the query network is set to $10^{-4}$ with the Adam optimizer Kingma & Ba (2014) while the learning rate of the synthesis network stays the same with the original methods. The batch size is 128, and the random seed is set to 100.

**List processing.** The query encoder is the same as the one in PCCoder. The dimension of $\boldsymbol{\mu}$ and $\boldsymbol{\sigma}$ is set to 256 for dataset $D_1$ and 128 for dataset $D_2$ without much tuning. The query decoder is a single-layer linear network with dimension 256 for dataset $D_1$ and 128 for dataset $D_2$. The program encoder is a single layer LSTM with hidden size 256 for dataset $D_1$ and 128 for dataset $D_2$.

The inputs of list processing consist of three types: INT, LIST, and NULL. Thus, each query is NULL or an integer or a list consisting of integers in the range of $[-256, 255]$. Each integer is represented by a $512$ one-hot vector. For the well-designed input-output examples, the length of a LIST is sampled stochastically with the maximum length of 20. However, to make the query network simpler, we fix the length of queries to 20. The query network generates all three types

$$
\begin{aligned}
\text{Prog } p \quad &:= \quad \texttt{def run()} : s \\
\text{Stmt } s \quad &:= \quad \texttt{while}(b) : s \mid \texttt{repeat}(r) : s \mid s_1 ; s_2 \mid a \\
&\quad \mid \quad \texttt{if}(b) : s \mid \texttt{ifelse}(b) : s_1 \texttt{ else} : s_2 \\
\text{Cond } b \quad &:= \quad \texttt{frontIsClear()} \mid \texttt{leftIsClear()} \mid \texttt{rightIsClear()} \\
&\quad \mid \quad \texttt{markersPresent()} \mid \texttt{noMarkersPresent()} \mid \texttt{not } b \\
\text{Action } a \quad &:= \quad \texttt{move()} \mid \texttt{turnRight()} \mid \texttt{turnLeft()} \\
&\quad \mid \quad \texttt{pickMarker()} \mid \texttt{putMarker()} \\
\text{Cste } r \quad &:= \quad 0 \mid 1 \mid \cdots \mid 19
\end{aligned}
$$

Figure 8: The DSL of Karel

separately by different networks and chooses among them according to the type of inputs using a type filter.

The details of the query network for list processing are shown in figure 7.

For training, the learning rate of the query network is set to $10^{-4}$ with a 0.1 decay every 40 epochs and the Adam optimizer Kingma & Ba (2014). The learning rate of the synthesis network is $10^{-3}$, which stays the same with the original methods with 0.1 decay every 4 epochs. The batch size of the query process is 64, the batch size of synthesis is 32 for $D_1$ and 100 for $D_2$. The random seed is set to 100.

### A.3   OTHER TRAINING TECHNIQUES

**Latent code.**   The subsequent queries are easy to fail into the mode collapse, generating similar queries repetitively. To tackle this problem, we introduce a latent code, like the one in Info-GAN (Chen et al., 2016), as another input, which indicates the current query step, and ask the encoder in the next query step to decode it accurately. Specifically, given a latent code $c$ which indicates the current query step, the query network is supposed to maximize the mutual information between $c$ and the output query $x$ to alleviate the mode collapse problem. To generate $x$ under the condition of $c$, we concatenate $c$ with the encoded representation $\mu$ and $log(\sigma^2)$, and then send it to the decoder (Equation (9)), shown in Figure 6(g) and Figure 7(e). To maximize the mutual information $I(c; x)$, we design a network aiming to classify $c$ from $x$, which models the distribution $Q(c|x)$ (details can be seen in Chen et al. (2016)). This network shares its parameters with the $Encoder_e$ (Equation (5)) except for the last layer, and it is optimized with the cross-entropy loss.

**Gumbel-Softmax.**   Note that the queries are discrete, which makes the query network cannot be optimized by back-propagation. Thus, we take advantage of the Gumbel-Softmax distribution (Maddison et al., 2017; Jang et al., 2017) and make the query process differentiable.

**Curriculum learning.**   The query network is trained progressively. At the beginning of the training, the query network generates one query only. As the training goes on, the limit of the number of queries increases until it achieves five. Specifically, this limit increases every two epochs. Curriculum learning helps to make the training process more stable.

**Kullback-Leibler (KL) divergence.**   A failed attempt. Similar to the latent code, we tried to add the reciprocal of the KL divergence between different queries on the same program as another loss to make the queries more diverse. However, this loss does not have a significant impact on training most of the time and makes the training process unstable, and thus it is abandoned in our final version.

| Hero facing North |
| --- |
| Hero facing South |
| Hero facing West |
| Hero facing East |
| Obstacle |
| Grid boundary |
| 1 marker |
| 2 marker |
| 3 marker |
| 4 marker |
| 5 marker |
| 6 marker |
| 7 marker |
| 8 marker |
| 9 marker |
| 10 marker |

Table 4: Representation of a cell in grid world

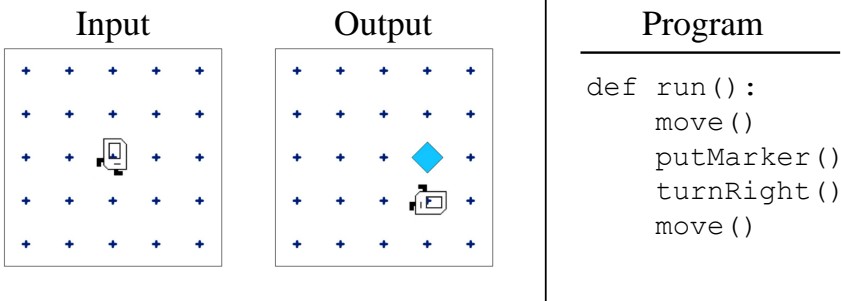

Figure 9: An example of Karel.

# B    EXPERIMENT DETAILS

## B.1    THE KAREL TASK

Karel is an educational programming language, which can be used to control a robot to move in a 2D grid world (Pattis et al., 1981). Figure 8 presents the domain language specification (DSL) of Karel. Figure 9 shows a Karel program example. Each cell of the grid world is represented as a 16-dimensional vector corresponding to the features described in Table 4 (Bunel et al., 2018).

**Handling of crashes.**    The executor of Karel will get a "CRASH" result and then terminates if the agent:

- Picks a marker while no marker is presented.
- Puts a marker down while the number of the markers in the cell exceeds the limit (the limit is set to 10).
- Walks up an obstacle or out of boundaries.
- Falls into an infinite loop (the loop with more than $10^5$ API calls).

In the early stage of training, the query network generates the queries randomly, and thus the programs run into these crashed states easily, making the queries cannot be applied to training. To avoid these situations, we modify the executor so that when crashes happen, the state stays still without change and the program keeps executing.

$$
\begin{aligned}
\text{Basic Function} \quad &:= \quad +1 \mid -1 \mid \times 2 \mid \div 2 \mid \times(-1) \\
& \mid \quad **2 \mid \times 3 \mid \div 3 \mid \times 4 \mid \div 4 \\
& \mid \quad > 0 \mid < 0 \mid \%2 \mid \%2 == 1 \\
\text{First-order Function} \quad &:= \quad \texttt{HEAD} \mid \texttt{LAST} \mid \texttt{TAKE} \mid \texttt{DROP} \mid \texttt{ACCESS} \\
& \mid \quad \texttt{MINIMUM} \mid \texttt{MAXIMUN} \mid \texttt{REVERSE} \mid \texttt{SORT} \mid \texttt{SUM} \\
\text{Higher-order Function} \quad &:= \quad \texttt{MAP} \mid \texttt{FILTER} \mid \texttt{COUNT} \mid \texttt{ZIPWITH} \mid \texttt{SCANL1}
\end{aligned}
$$

Figure 10: The DSL of list processing

| Input | Program |
|---|---|
| [-17, -3, 4, 11, 0, -5, -9, 13, 6, 6, -8, 11] | FILTER (<0) |
| **Output** | MAP (×4) |
| [-12, -20, -32, -36, -68] | SORT |
| | REVERSE |

Figure 11: An example of list processing.

## B.2 THE LIST PROCESSING TASK

The list processing task takes 1-3 lists or integers as input examples and produces a list or an integer as the output example. An example is shown in Figure 11. Figure 10 shows the DSL of list processing. Following Zohar & Wolf (2018), we generate two datasets: $D_1$ with program length 4 and $D_2$ with program length up to 12.

**Handling of the out of range problem.** Constraint propagation guarantees that the execution of programs never obtains intermediate values that are out of range $[-256, 255]$. However, when we query randomly in the early stage of training, the out-of-range problem can easily occur. To handle this problem, we truncate the intermediate values to $[-256, 255]$ while querying to ensure that all queries can yield legal responses.

## B.3 THE EVOLUTION OF DISTRIBUTION ENTROPY

To make a sanity check of the F-space formulation, we study how the entropy of the distribution changes over the step of the query.

The entropy of a multivariate Normal distribution $\mathcal{N}(x; \mu, \Sigma))$ is given by

$$
H(x) = \frac{1}{2}ln|\Sigma| + \frac{D}{2}(1 + ln(2\pi)), \tag{11}
$$

where $\Sigma$ denotes the covariance matrix and $D$ denotes the dimension of $x$. In our case, we have assumed the independence of each dimension of $x$ which means that $\Sigma$ is a diagonal matrix. Hence

$$
\frac{1}{2}ln(|\Sigma|) = \frac{1}{2}ln(\prod_i \sigma_i^2) = \frac{1}{2}\sum_i ln(\sigma_i^2), \tag{12}
$$

where $\sigma_i^2$ is the diagonal element of $\Sigma$, indicating the variance of each dimension. $D$ is a constant during querying, so we calculate the mean of $log(\boldsymbol{\sigma}^2)$ as an equivalent substitute to show the change of the entropy. Results are shown in Figure 12. In our experiments, Karel performs best and this is also revealed by the entropy that Karel decreases much faster than list processing. On the contrary, the entropy of list processing decreases slowly and has worse performance in our experiment. This performance may be increased by tuning the query network more carefully.

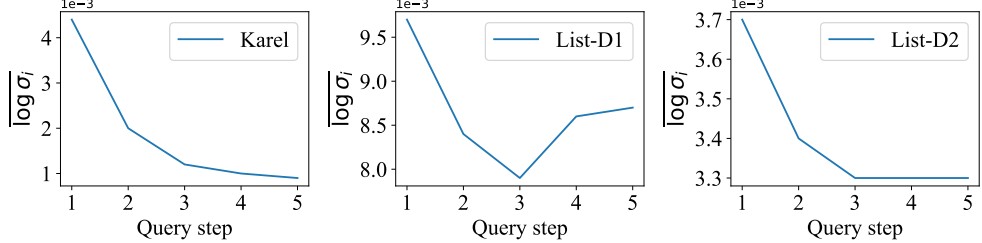

Figure 12: The entropy of the distribution decays when query goes on.

Table 5: The statistics on the query times of each crash-aware QBC query step.

|  | 1 | 2 | 3 | 4 | 5 |
|---|---|---|---|---|---|
| avg | 5.72 | 2.47 | 2.96 | 4.27 | 4.86 |
| max | 214 | 200 | 224 | 357 | 226 |
| min | 0 | 0 | 0 | 0 | 0 |

### B.4 QUERY BY COMMITTEE

In this section, we present the details of our baseline algorithm: query by committee (Seung et al., 1992). Algorithm 2 shows the crash-aware version of the QBC algorithm, which selects the query that can result in the most diverse outputs. The crash-unaware version can be obtained simply by removing all CRASH judgments. Note that, compared with the crash-unaware version, the crash-aware version queries the underlying program more times for CRASH judgment (see Table 5), and thus results in a much better performance.

## C THEOREMS AND PROOFS

### C.1 THE PRODUCT OF TWO NORMAL DISTRIBUTIONS

In this section, we will show that the product of two normal distributions is a scaled normal distribution.

**Theorem C.1** (The product of two normal distributions). *Given two normal distributions that satisfies* $p(x) = \frac{1}{\sqrt{2\pi}\sigma}e^{-\frac{(x-\mu)^2}{2\sigma^2}}$, *the product of them is a scaled normal "distribution" in the form of* $\alpha\frac{1}{\sqrt{2\pi}\sigma'}e^{-\frac{(x-\mu')^2}{2\sigma'^2}}$, *where* $\alpha$ *is the scale factor.*

*Proof.* Suppose we have two normal distributions $p_a$ and $p_b$:

$$p_a(x) = \frac{1}{\sqrt{2\pi}\sigma_a}e^{-\frac{(x-\mu_a)^2}{2\sigma_a^2}},$$
$$p_b(x) = \frac{1}{\sqrt{2\pi}\sigma_b}e^{-\frac{(x-\mu_b)^2}{2\sigma_b^2}}.$$

$$(13)$$

The product of $p_a$ and $p_b$:

$$p_a(x)p_b(x) = \frac{1}{\sqrt{2\pi}\sigma_a}e^{-\frac{(x-\mu_a)^2}{2\sigma_a^2}} \cdot \frac{1}{\sqrt{2\pi}\sigma_b}e^{-\frac{(x-\mu_b)^2}{2\sigma_b^2}}$$
$$= \frac{1}{2\pi\sigma_a\sigma_b}e^{-\left(\frac{(x-\mu_a)^2}{2\sigma_a^2} + \frac{(x-\mu_b)^2}{2\sigma_b^2}\right)}.$$

$$(14)$$

---

**Algorithm 2** Query by committee (QBC): crash-aware

---

1: **function** QUERY( )
2:     Initialize trained model $M$, query pool $Q$ and max query times $T$
3:     $p \leftarrow Underlying\ program\ (oracle)$
4:     **repeat**
5:         $x_1 \leftarrow sample\,from\,Q$
6:         $y_1 \leftarrow p(x_1)$
7:     **until** $y_1$ not CRASH
8:     $[\![e]\!] \leftarrow \{(x_1, y_1)\}$
9:     **for** $t \in \{2 \dots T\}$ **do**
10:         program candidates $[\![\hat{p}]\!] \leftarrow M([\![e]\!])$          $\triangleright$ Get top K predictions by beam search
11:         $x_t \leftarrow$ SELECT-QUERY$(Q, p, [\![\hat{p}]\!])$
12:         $y_t \leftarrow p(x_t)$
13:         $[\![e]\!] \leftarrow [\![e]\!] \cup \{(x_t, y_t)\}$
14:     **end for**
15:     **return** $[\![e]\!]$
16: **end function**

1: **function** SELECT-QUERY$(Q, p, [\![\hat{p}]\!])$
2:     **repeat**
3:         $queries \leftarrow$ sample 100 times from $Q$
4:         $score\_list \leftarrow []$          $\triangleright$ Acquisition scores of queries
5:         **for** $q \in queries$ **do**
6:             $s_q \leftarrow 0$
7:             **for** $\hat{p} \in [\![\hat{p}]\!]$ **do**
8:                 $\hat{y} \leftarrow \hat{p}(q)$
9:                 **if** $\hat{y}$ is unique **then**
10:                     $s_q \leftarrow s_q + 1$          $\triangleright$ The more diversity the output, the better the query
11:                 **end if**
12:             **end for**
13:             $score\_list.append((q, s_q))$
14:         **end for**
15:         Sort $score\_list$ in a descending order with $s_q$
16:         **for** $q \in score\_list$ **do**      $\triangleright$ Select the query with the highest score and without CRASH
17:             $y \leftarrow p(q)$
18:             **if** $y$ not CRASH **then**
19:                 **return** $q$
20:             **end if**
21:         **end for**
22:     **until** a query is found          $\triangleright$ If all 100 queries result in CRASH, repeat this process.
23: **end function**

---

Consider the index part

$$
\begin{aligned}
\frac{(x - \mu_a)^2}{2\sigma_a^2} + \frac{(x - \mu_b)^2}{2\sigma_b^2} &= \frac{(\sigma_a^2 + \sigma_b^2)x^2 - 2(\mu_b\sigma_a^2 + \mu_a\sigma_b^2)x + (\mu_a^2\sigma_b^2 + \mu_b^2\sigma_a^2)}{2\sigma_a^2 2\sigma_b^2} \\
&= \frac{x^2 - 2\frac{\mu_b\sigma_a^2 + \mu_a\sigma_b^2}{\sigma_a^2 + \sigma_b^2}x + \frac{\mu_b^2\sigma_a^2 + \mu_a^2\sigma_b^2}{\sigma_a^2 + \sigma_b^2}}{\frac{2\sigma_a^2\sigma_b^2}{\sigma_a^2 + \sigma_b^2}} \\
&= \frac{(x - \frac{\mu_b\sigma_a^2 + \mu_a\sigma_b^2}{\sigma_a^2 + \sigma_b^2})^2}{\frac{2\sigma_a^2\sigma_b^2}{\sigma_a^2 + \sigma_b^2}} + \frac{\frac{\mu_b^2\sigma_a^2 + \mu_a^2\sigma_b^2}{\sigma_a^2 + \sigma_b^2} - (\frac{\mu_b\sigma_a^2 + \mu_a\sigma_b^2}{\sigma_a^2 + \sigma_b^2})^2}{\frac{2\sigma_a^2\sigma_b^2}{\sigma_a^2 + \sigma_b^2}} \\
&= \gamma + \lambda,
\end{aligned}
\tag{15}
$$

where

$$\gamma = \frac{(x - \frac{\mu_b \sigma_a^2 + \mu_a \sigma_b^2}{\sigma_a^2 + \sigma_b^2})^2}{\frac{2\sigma_a^2 \sigma_b^2}{\sigma_a^2 + \sigma_b^2}},$$

$$\lambda = \frac{\frac{\mu_b^2 \sigma_a^2 + \mu_a^2 \sigma_b^2}{\sigma_a^2 + \sigma_b^2} - (\frac{\mu_b \sigma_a^2 + \mu_a \sigma_b^2}{\sigma_a^2 + \sigma_b^2})^2}{\frac{2\sigma_a^2 \sigma_b^2}{\sigma_a^2 + \sigma_b^2}}. \tag{16}$$

Simplify $\lambda$:

$$\begin{aligned}
\lambda &= \frac{\frac{\mu_b^2 \sigma_a^2 + \mu_a^2 \sigma_b^2}{\sigma_a^2 + \sigma_b^2} - (\frac{\mu_b \sigma_a^2 + \mu_a \sigma_b^2}{\sigma_a^2 + \sigma_b^2})^2}{\frac{2\sigma_a^2 \sigma_b^2}{\sigma_a^2 + \sigma_b^2}} \\
&= \frac{(\mu_b^2 \sigma_a^2 + \mu_a^2 \sigma_b^2)(\sigma_a^2 + \sigma_b^2) - (\mu_b \sigma_a^2 + \mu_a \sigma_b^2)^2}{2\sigma_a^2 \sigma_b^2 (\sigma_a^2 + \sigma_b^2)} \\
&= \frac{\sigma_a^2 \sigma_b^2 (\mu_a^2 + \mu_b^2 - 2\mu_a \mu_b)}{2\sigma_a^2 \sigma_b^2 (\sigma_a^2 + \sigma_b^2)} \\
&= \frac{(\mu_a - \mu_b)^2}{2(\sigma_a^2 + \sigma_b^2)}.
\end{aligned} \tag{17}$$

Finally, we get

$$\begin{aligned}
p_a(x)p_b(x) &= \frac{1}{2\pi \sigma_a \sigma_b} e^{-\lambda} e^{-\gamma} \\
&= \alpha \cdot \frac{1}{\sqrt{2\pi}\sigma'} e^{-\frac{(x - \mu')^2}{2\sigma'^2}},
\end{aligned} \tag{18}$$

where

$$\begin{aligned}
\mu' &= \frac{\mu_b \sigma_a^2 + \mu_a \sigma_b^2}{\sigma_a^2 + \sigma_b^2}, \\
\sigma' &= \sqrt{\frac{\sigma_a^2 \sigma_b^2}{\sigma_a^2 + \sigma_b^2}}, \\
\alpha &= \frac{1}{\sqrt{2\pi(\sigma_a^2 + \sigma_b^2)}} e^{-\frac{(\mu_a - \mu_b)^2}{2(\sigma_a^2 + \sigma_b^2)}}.
\end{aligned} \tag{19}$$

$\square$

Next, we will show that approximating the new $\mu'$ and $log(\sigma'^2)$ with the weighted sum of the $[\mu_a, \mu_b]$ and $[log(\sigma_a^2), log(\sigma_b^2)]$ is reasonable.

For $\mu'$, it is obvious that it can be seen as

$$\begin{aligned}
\mu' &= \frac{\mu_b \sigma_a^2 + \mu_a \sigma_b^2}{\sigma_a^2 + \sigma_b^2}, \\
&= \frac{\sigma_b^2}{\sigma_a^2 + \sigma_b^2} \cdot \mu_a + \frac{\sigma_a^2}{\sigma_a^2 + \sigma_b^2} \cdot \mu_b,
\end{aligned} \tag{20}$$

which is a weighted sum. For $log(\sigma'^2)$, we can rewrite it as follows

$$\begin{aligned}
log(\sigma'^2) &= log(\frac{\sigma_a^2 \sigma_b^2}{\sigma_a^2 + \sigma_b^2}), \\
&= log(\sigma_a^2) + log(\sigma_b^2) - log(\sigma_a^2 + \sigma_b^2) \\
&= log(\sigma_a^2) + log(\sigma_b^2) - (h\,log(\sigma_a^2) + k\,log(\sigma_b^2) + log(\frac{1}{\sigma_b^{2k}\sigma_a^{2(h-1)}} + \frac{1}{\sigma_a^{2h}\sigma_b^{2(k-1)}})) \\
&= (1 - h)log(\sigma_a^2) + (1 - k)log(\sigma_b^2) - log(\frac{1}{\sigma_b^{2k}\sigma_a^{2(h-1)}} + \frac{1}{\sigma_a^{2h}\sigma_b^{2(k-1)}}),
\end{aligned} \tag{21}$$

| program | $q_A$ | $q_B$ | $q_C$ | $q_D$ |
|---------|-------|-------|-------|-------|
| $p_0$ | $\checkmark$ | $\times$ | $\times$ | $\times$ |
| $p_1$ | $\checkmark$ | $\times$ | $\times$ | $\checkmark$ |
| $p_2$ | $\checkmark$ | $\times$ | $\checkmark$ | $\checkmark$ |
| $p_3$ | $\checkmark$ | $\checkmark$ | $\checkmark$ | $\times$ |
| $p_4$ | $\times$ | $\checkmark$ | $\checkmark$ | $\times$ |
| $p_5$ | $\times$ | $\checkmark$ | $\checkmark$ | $\checkmark$ |
| $p_6$ | $\times$ | $\checkmark$ | $\times$ | $\times$ |
| $p_7$ | $\times$ | $\times$ | $\times$ | $\checkmark$ |

Figure 13: An example: 8 candidate programs $p_{0-7}$ and 4 queries $q_{A-D}$ each with 2 possible responses $\{\checkmark, \times\}$.

where $h$ and $k$ are two coefficients. With suitable $h$ and $k$ learned, the last term can be ignored ($\frac{1}{\sigma_b^{2k}\sigma_a^{2(h-1)}} + \frac{1}{\sigma_a^{2h}\sigma_b^{2(k-1)}} \approx 1$) and thus $log(\sigma'^2)$ can be approximate by the weighted sum of $[log(\sigma_a^2), log(\sigma_b^2)]$.

### C.2 THE RECURRENT TRAINING PROCESS

We adopt a recurrent training process to model the mutual information between the input-output examples and the programs more accurately. Traditionally, the next query is gained by selecting the query with the maximum mutual information conditioned on the former ones:

$$q_k = \arg\max_x I(\mathbb{P}; (x, y)|[\![e]\!]_{k-1}) \tag{22}$$

However, this greedy strategy fails in some cases. An example is shown in Figure 13. Suppose that after a series of queries, and finally there are only 8 candidate programs $\mathbb{P} = \{p_0, ..., p_7\}$ distributed uniformly and 4 queries $\mathbb{Q} = \{q_A, ..., q_D\}$ each with 2 possible responses $\{\checkmark, \times\}$ left, our goal is to find out the underlying program with as few queries as possible (conditioned on the former queries which are omitted). According to the definition of the mutual information $I(X; Y) = \sum_{x \in X} \sum_{y \in Y} P(x, y) log \frac{P(x,y)}{P(x)P(y)}$, we can calculate the mutual information between the 4 queries and the programs as follows (we use $q_{A-D} = \checkmark//\times$ to represent $p_i(q_{A-D}) = \checkmark//\times$ for simplicity):

$$I(\mathbb{P}; q_A) = \sum_{p \in \mathbb{P}} \sum_{q_A \in \{\checkmark, \times\}} P(p, q_A) log \frac{P(p, q_A)}{P(p)P(q_A)} \quad = 8 * \frac{1}{8} * log(\frac{\frac{1}{8}}{\frac{1}{8} * \frac{1}{2}}) = 1. \tag{23}$$

Samilarly, $I(\mathbb{P}; q_B) = I(\mathbb{P}; q_C) = I(\mathbb{P}; q_D) = 1$. Thus, Equation 22 suggests that 4 queries share the same priority. When the query process continues, however, this is not the case. For the second query (let $q = (q_A, q_B)$):

$$I(\mathbb{P}; q) = \sum_{p \in \mathbb{P}} \sum_{q \in \{\checkmark, \times\}^2} P(p, q) log \frac{P(p, q)}{P(p)P(q)} \quad = 6 * \frac{1}{8} * log(\frac{\frac{1}{8}}{\frac{1}{8} * \frac{3}{8}}) + 2 * \frac{1}{8} * log(\frac{\frac{1}{8}}{\frac{1}{8} * \frac{1}{8}}) = 1.81, \tag{24}$$

where $\{\checkmark, \times\}^2 = \{\checkmark, \times\} \times \{\checkmark, \times\}$. Similarly,

$$\begin{aligned} I(\mathbb{P}; (q_A, q_C)) = I(\mathbb{P}; (q_A, q_D)) = I(\mathbb{P}; (q_C, q_D)) = 2, \\ I(\mathbb{P}; (q_B, q_C)) = I(\mathbb{P}; (q_B, q_D)) = 1.81. \end{aligned} \tag{25}$$

Equation 24 and Equation 25 show that considering the longer horizon, $q_B$ is the worst choice as the first query because it gets the minimum mutual information whichever the second query is. This conclusion is also straightforward without calculation: if we choose $q_{A,C,D}$ as the first query, then we only need to query for two more times; And if we choose $q_B$ as the first query, then we need 3.25 queries on average with 4 queries in the worst case.

To this end, we adopt a recurrent training process as shown in Figure 3. In the recurrent training process, the gradient can be propagated through multiple query steps and thus the current query selection can be affected by future queries.

## D   ADDITIONAL RELATED WORK

### D.1   ACTIVE LEARNING

Active learning is a research domain that aims to reduce the cost of labeling by selecting the most representative samples iteratively from the unlabeled dataset and then asking them to an oracle for labeling while training. According to Shui et al. (2020), active learning can be divided into pool-based sampling (Angluin, 1988; King et al., 2004), which judges whether a sample should be selected for query-based on the evaluation of the entire dataset; steam-based sampling (Dagan & Argamon, 1995; Krishnamurthy, 2002), which judges each sample independently compared to pool-based sampling; and membership query synthesis (Lewis & Gale, 1994), which means that the unlabeled sample can be generated by the learner instead of selecting from the dataset only. Although active learning involves querying an oracle iteratively, which is similar to our framework, there are still two main differences between them. (1) For the final purpose, active learning aims to finish the training stage at a low cost, and the inference stage remains the same as other machine learning tasks without the query process, while our framework aims to find the oracle (*i.e.*the underlying program) in a symbolic form, and the query process is retained during inference. (2) For the query process, active learning assumes only one oracle (*i.e.*the same query always gets the same label). In contrast, our framework assumes multiple oracles (*i.e.*the same query will get different responses if the underlying programs are different).

### D.2   AUTOMATED BLACK-BOX TESTING

Black-box testing is a method of software testing aiming to examine the functionality of a black-box (such as a piece of a program) by a large number of test cases without perceiving its internal structures. The most famous automated test cases generation method is learning-based testing (LBT) (Meinke, 2004; Meinke & Niu, 2010; Meinke & Sindhu, 2011). LBT is an iterative approach to generate test cases by interacting with the black-box, which sounds similar to the query problem mentioned in this paper. However, the settings and the purpose of the black-box testing are totally different from ours. In detail, the black-box testing assumes the existence of a target requirement (or target function) and a black-box implementation of this requirement (such as a piece of program which is unknown). The purpose is to check the black-box implementation to ensure that there is no bug or difference between the target requirement and the black-box by generating test cases (*i.e.*input-output examples). In a contrast, in our settings, the target function does not exist, and all we have is a black-box (or called oracle / underlying program in our paper). Our goal is to query this black box and guess which program is hidden inside based on the experience learned from the training set.

## E   TWO QUERY EXAMPLES

In this section, we show two query examples that cover all branches of the program while the well-designed dataset fails to do this. Each example consists of the underlying program and the corresponding important queries that improve the branch coverage. See Figure 14 and Figure 15.

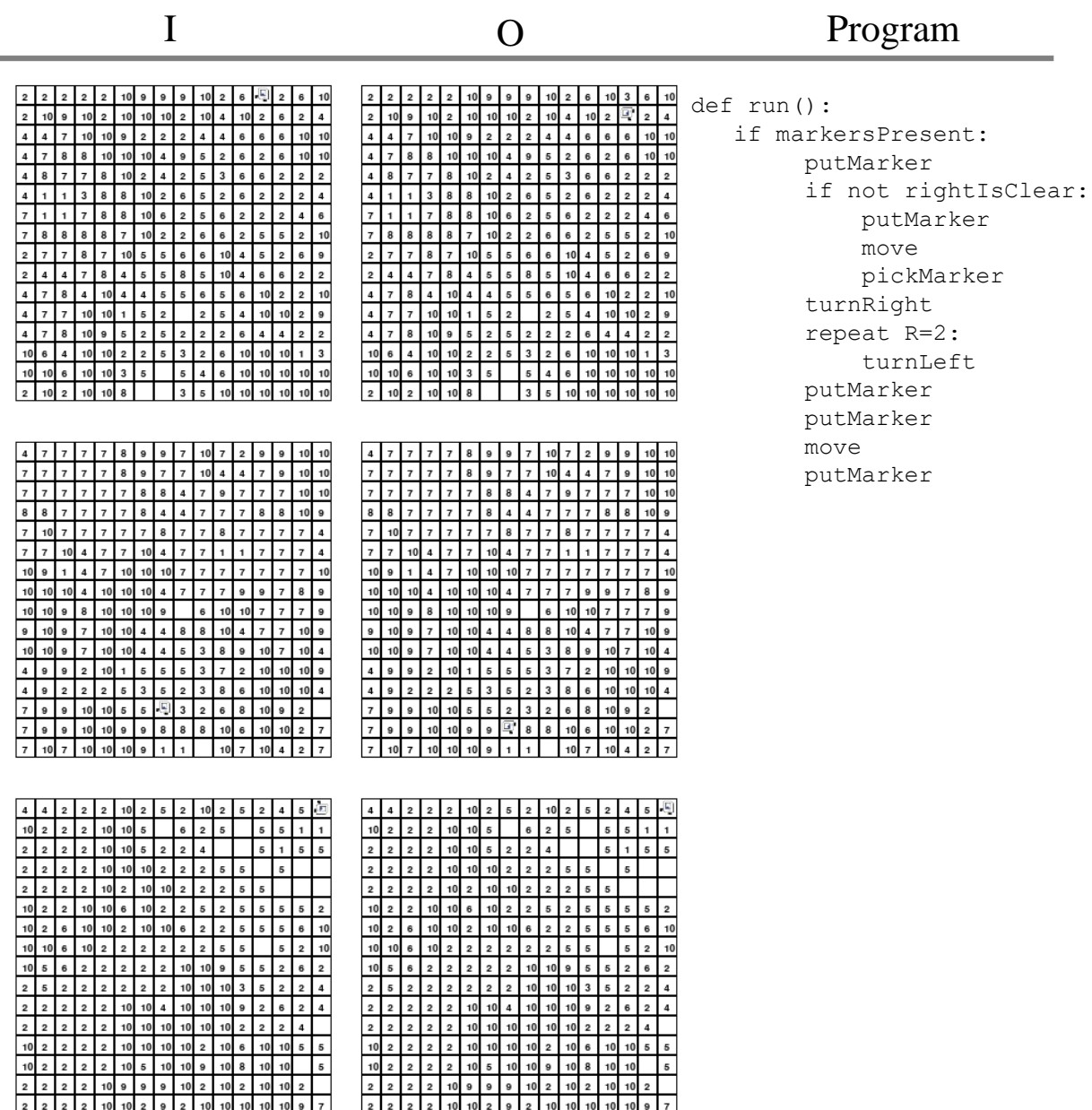

Figure 14: An query example of Karel. The number in the cell denotes the amount of markers.

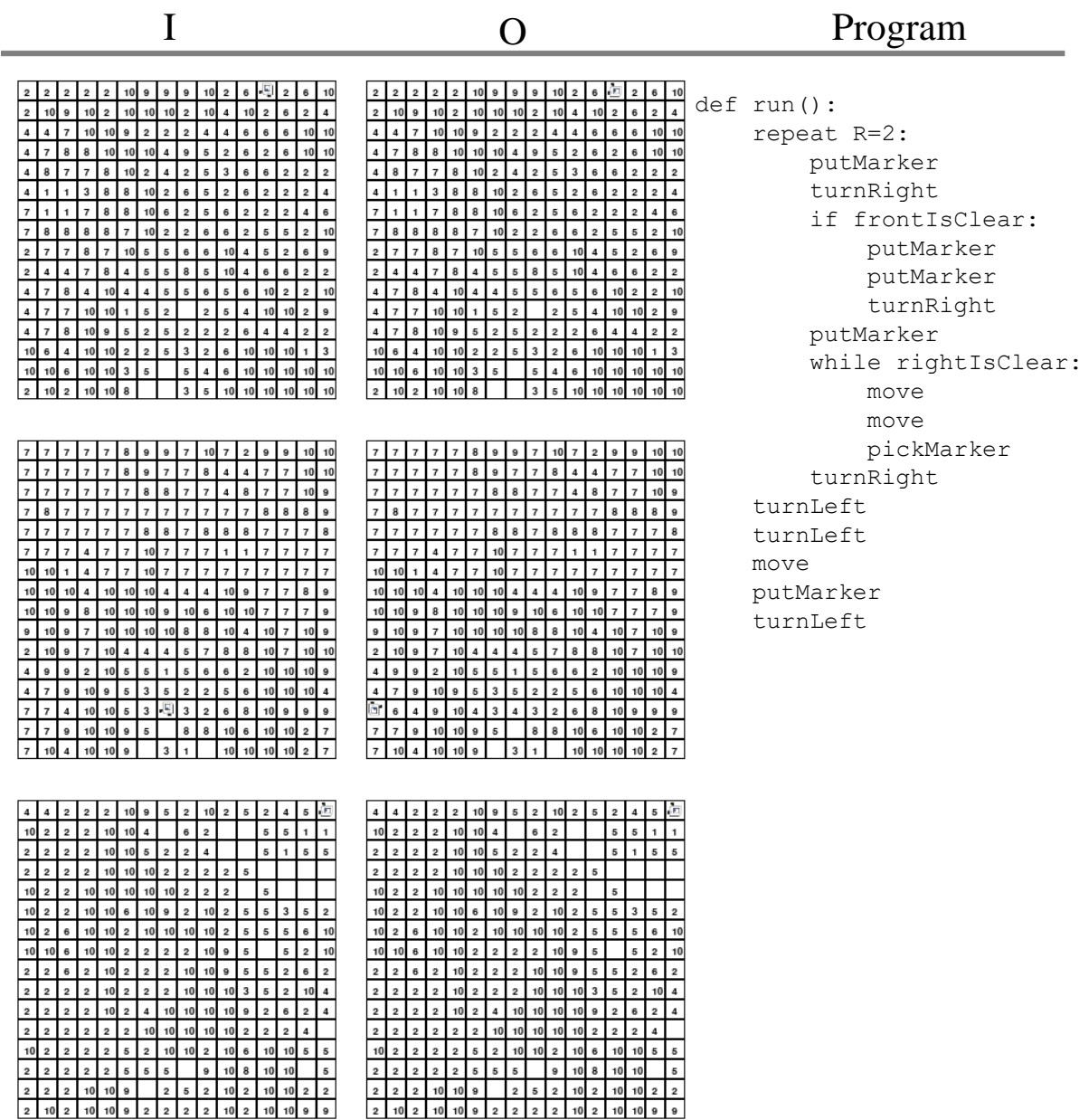

Figure 15: Another query example of Karel. The number in the cell denotes the amount of markers.

