# OpenReview forum: "Neural Program Synthesis with Query"
_ICLR.cc/2022/Conference — ICLR 2022 Poster_

### Official Review · Reviewer_WAY8 · 2021-10-26

**Correctness:** 4
**Technical Novelty And Significance:** 3
**Empirical Novelty And Significance:** 3
**Recommendation:** 8
**Confidence:** 4

**Main Review:**

## what I like about this paper

I really liked the F-space formulation, it is very clever and intuitive to map single functions p to points, and sets of examples e to distribution of points, it is a "by definition" representation that captures the relationships between examples and programs in a learned embedding. the operations over this space is also well-justified, with algebraic manipulations of the example embedding such as union and intersections very cleanly explained.

Algorithm 1. is cleanly written, I was able to follow it well other than line 11 and 12 (see question below)

I personally liked the choice of functionally equivalent proxy for evaluation. "if the predicted program satisfies all 100 examples, then it is functional equivalent to the ground-truth program". It is a very practical choice and and also a fairly thorough one.

The results seem convincing to me, implementing a query generator for the space of Karel and list-manipulation is very impressive, as these are fairly high-dimension spaces, and much work must have been put into making it all work out.

Overall this is a solid work that (appears to me) is written by a group who understands the landscapes of neuro-guided program synthesis well.

## what this paper needs work on
This paper would be much stronger if the following points can be addressed.

### the exposition
while the technical aspect of the paper (F-space, its learning objective, evaluation) are sound, the problem statement can be cleaner. the current paper does not draw a clean distinction between "the problem statement" and "the proposed algorithm". as a result, the paper reads like a long todo list of "we need to do this, then did that, oh by the way we need to make this differentiable because neural networks" and as a reader I'm left confused as to what the actual problem is that we're solving. for instance, in definition 2.3, it tells me what a querying algorithm is doing, but it doesn't tell me at all _why_ we need to query or _what_ are we querying for.

to be constructive, I would add a "problem statement" subsection before "problem formulation" that formally define what exactly is the problem we're solving. specifically, one might borrow the terminologies of active diagnostics / bayesian optimization, that of the "acquisition function": given a set of possible query points X, the acquisition function a(x) scores all possible query points x \in X, and the querying algorithm picks the x with the highest score. in this particular work, the acquisition function is that of mutual information I([[e]], p), which is to say, given a set of (potentially empty) observed examples so-far {(x1,y1)...(xt-1,yt-1)}, we wish to have the query network pick a next query point x_t such that the acquisition function of mutual information I([[e + (xt, yt)]], p) is maximized. _only then_ can one jump into the explanation of F-space and related techniques of solving this querying problem.

currently, the relationship to mutual information (the objective that the query network is optimizing for) is only mentioned in equation (7), almost as an after-thought. this is incredibly confusing because I am left clueless what problem the paper is actually solving while reading the answers to this mysterious problem (think hitchhiker's guide, where the answer is 42 but the question is unstated).

Since we use the same acquisition function both in training the synthesizer and at inference time to query, the synthesizer receives the examples from the same distribution, thus, we do not run into the problem of out-of-distribution examples at inference time. The paper definitely makes an attempt to make this claim, but it is unclear in the current version. I believe once the problem statement is cleanly defined, this contribution can be stated cleaner as well.

### other baselines
Current approach only compares against a weak random baseline, while there is merits to it (one can argue random examples is the standard for synthesis benchmarks), the paper can be much stronger with these additional baselines:

1) query by committee. this is a standard approach in active diagnostics where one keeps a sampled distribution of valid hypothesis, then select the query point that maximizes disagreement. specifically, given current set of examples e = {(x1,y1)...(xk,yk)}, we can sample the a program synthesizer to obtain multiple programs that are consistent with e: p1, p2, ... pn, then, use 100 randomly sampled inputs (same as your testing procedure) to discover if there is an input x that maximizes disagreement between these programs. Terminate early of the synthesizer can only propose functionally equivalent programs over the 100 sampled inputs. See https://dl.acm.org/doi/pdf/10.1145/130385.130417 for a similar set up, albeit over a simpler domain.

2) end-to-end RL. one can train a query proposer jointly with a synthesizer in an end-to-end communication setting, where one jointly optimizes (over both the query NN and the synthesizer NN) the probability of the synthesizer recovering the ground-truth program p, given the observation sampled by the querying NN (x, p(x))
See https://arxiv.org/pdf/2006.00418.pdf for an end-to-end communication game set up, albeit over a simpler domain. I believe the same gumbel trick would work here as well.

I would expect a replication of 1), as it is by far the most standard approach in active diagnostics, and is incredibly easy to implement and used widely in practice. 2) might be more difficult to replicate, I am okay not seeing it. I believe 1) and 2) serve as strong, yet standard baselines that this work should compare itself to, and I am fairly confident the proposed method of this work would out-perform both 1) and 2), seeing the proposed approach demonstratively out-performs 1) and 2) will make me much more confident in this work.

### related works

There are very relevant prior works that studies the relationships between optimally querying examples for program synthesis from an information theoretic perspective:
[1] https://arxiv.org/pdf/1704.06131.pdf
[2] https://arxiv.org/pdf/1711.03243.pdf

Specifically, these works formally justify the strategy of greedily picking the next query (i.e. picking x_t that greedily "maximize the mutual information between the input-output examples [[e]] and the corresponding program p", same as this work) as more than a heuristic, but is in fact (1-1/e) as good as the globally optimal solution that picks (globally) a set of k examples that maximize the mutual information between [[e]] and p. The proof involves showing that adding examples is monotonic and sub-modular in reducing the set of satisfiable programs.

Further, [1] studies the exact same problem as this work: How do you optimally query a set of observations for program synthesis? And also propose a similar querying NN that selects which next point to add, however, unlike this work, they assume the space of query points can be enumerated. Thus, one can view this work as solving a much needed algorithmic improvement to [1], by making a querying NN that can _generate_ new query points. Explaining the relationship between this work and [1,2] would help contextualizing this work better.

## questions:

- line 11 and 12 of algorithm 1 is bit unclear to me. The decoder makes a choice for the next query point x_t, but since the rest of the loss requires one to query the oracle p(x_t), the gradient cannot "go through" this sample step. typically this is learned using policy gradient or some RL objective, because presumably we cannot teacher force the correct x_t, because it amounts to sampling, from the space of all possible query points X, the best one that helps the identification of p. how is this done? (edit: ah gumbel trick. very nice!)

- how do you obtain negative samples? i.e. " we construct positive pairs {([[e]], p_i)} and negative pairs {([[e]], p_j )|i != j}. " how does this work? presumably you want to sample p_j in such a way that p_i and p_j are functionally equivalent over [[e]] yet distinct over some other query points (i.e. exactly the x your query network should produce). however sampling this p_j that is consistent to p_i sofar over [[e]] yet still distinct to some unseen x is a challenge of its own, so you adopted to just sample a "random p_j" if I understand correctly. is this the case? if so it probably should be justified, at least with a few words why this particular form of contrastive learning is a good choice here.

- is the synthesizer trained completely de-coupled from the latent representations of the querying NN? I'd imagine the embedding [[e]] be quite helpful, you just have to decode from it to obtain the correct program. but it is good either way, I would personally find the paper stronger if the synthesizer is completely decoupled, i.e. does not rely on [[e]] in any way just to prove a point that one could use any "back-end" synthesizer they want.

minor:

- in abstract, what does the phrase "actual distribution" mean? the distribution that end-user would use to communicate their intent? a distribution of input that would allow the program to be uniquely identified from all the space of programs unambiguously? or is it just "the training distribution", which is what I believe it to be after reading the intro.





**Summary Of The Paper:**

This paper considers an active diagnostic setup for program synthesis as follows: There is an oracle function/program p* belonging to a space of functions, and at inference time, one would like to uncover p*. To do so, a querying algorithm iteratively/adaptively query the oracle with inputs x1, x2, ... xk, receiving outcomes y1, y2, ... yk in return. A synthesizer (or any inverse model capable mapping observations to hypothesis, doesn't really matter) then proposes a guess p' based on the collected query-observation pairs (called examples) {(x1,y1)...(xk,yk)}. One "wins the game" if p* and p` behave the same way (functional equivalent) on all inputs, using as few queries (i.e. small k) as possible.

To do this, the paper introduce a learned F-space, which both functions f and sets of examples e can map into. While functions map (using a learned embedding) to single points in F-space (each point in F-space represent a unique function), examples map (using yet another learned embedding) to distributions in F-space (each example set represent a _distribution of_ functions that are consistent with it). The querying network is trained to generate the next observation x conditioned on the existing observations {(x1,y1)...(xk,yk)} in such a way that maximally reduce the uncertainty of the F-space once the new example, (x, p*(x)) is augmented to the existing observations.

Results are convincing to me.

**Summary Of The Review:**

Overall I liked the paper, as it solves 2 key challenges of program synthesis:
1. The training distribution of the synthesizer does not match the testing distribution
2. One would like to recover the target program with as few examples as possible

It solves both problem by "mocking" the active diagnostic process of (2) in training time, having a trained querying network to propose the most informative queries. To build a querying network, the proposed method embed both programs and examples in a joint F-space, and use contrastive learning to train the querying NN to propose query points that maximally distinguish the target program p from distractor programs p'. The construction of F-space is both intuitive and algebraically elegant.

Results give good evidence of the proposed method as well.

This paper would be stronger if the 3 points mentioned: exposition, other baselines, and related works can be addressed, I am willing to raise the score accordingly.

The author addressed the 3 points mentioned, so I am raising the score as I said I would. Although there is no score of 7, I meant a 7 instead of an 8 as the baseline (QBC) is still fairly close to the proposed method, and is derived in much less time (a week).

---

> ### Author Response · Authors · 2021-11-18
> **Response to Reviewer WAY8 (1/2)**
>
> We thank the reviewer for the comments and suggestions to improve our paper.
>
> ### The exposition
>
> Thank you for your suggestion, we have revised the “problem formulation” part in our paper to illustrate our insight and theoretical basis better. In short, we supplement a “problem statement” section before “problem formulation” to give an intuition of the query problem and explain what should we query. Specifically, we give an intuition of the query problem at the beginning of the problem statement to make readers clearer about our problem settings. Then, we introduce the concept of acquisition function and mutual information, and explain the guidance based on which we choose the queries. More details can be seen in Section 2.1.
>
> ### Other baselines
>
> (1) Query by committee
>
> We conduct an experiment on the query committee as our new baseline method.
>
> Exact match:
>
> | Query step        | 1      | 2      | 3      | 4      | 5      |
> | ----------------- | ------ | ------ | ------ | ------ | ------ |
> | Query (ours)            | 23.52% | 37.48% | 38.08% | 38.16% | 41.12% |
> | Random            | 15.80% | 22.40% | 26.52% | 28.40% | 30.12% |
> | QBC-crash-aware   | 29.75% | 34.19% | 36.06% | 36.38% | 36.34% |
> | QBC-crash-unaware | 11.43% | 12.86% | 11.98% | 7.90%  | 6.55%  |
>
> Functional equivalence:
>
> | Query step        | 1      | 2      | 3      | 4      | 5      |
> | ----------------- | ------ | ------ | ------ | ------ | ------ |
> | Query (ours)            | 27.16% | 42.28% | 43.04% | 43.08% | 46.64% |
> | Random            | 18.16% | 25.16% | 29.52% | 31.68% | 34.00% |
> | QBC-crash-aware   | 34.46% | 40.54% | 43.85% | 44.97% | 44.97% |
> | QBC-crash-unaware | 13.33% | 20.48% | 23.61% | 21.35% | 21.07% |
>
> Explanation:
>
> We trained a program synthesizer the same as Bunel et. al[1] except that it can receive 1-5 IOs instead of 5 IOs only. The query by committee algorithm executes as follows: given {(x_1, y_1), …, (x_{t-1}, y_{t-1})}, we use the synthesizer to generate top-100 programs by beam search. Then, we select 100 inputs from the training set randomly (just take it as an input generator which will not suffer the out-of-distribution problem) and feed them into 100 programs to get 100x100 outputs. We select the input that has the most diversity (the more different outputs given on the 100 programs, the higher the diversity) as the next query x_t. The first query is selected randomly from the training set without scoring.
>
> In program synthesis, illegal queries will cause the crash (or out-of-range) problem that the program may terminate and return an error (See Appendix B.1 and B.2). So there are two strategies based on QBC: Crash-aware, where if the query crashes then repeat the algorithm to sample another one; And crash-unaware, which samples queries regardless of the crash problem. QBC-crash-unaware performs worse than Random because Random is chosen by filtering crashed IOs while QBC-crash-unaware may contain crashes. QBC-crash-aware performs much better than QBC-crash-unaware because it queries the underlying program multiple times to make sure that the query will not result in a crash, which is unfair. Even though, our method still outperforms QBC-crash-aware, which shows its advantage.
>
> We have supplemented all results to Experiment 4.2 " Comparison with others" and Figure 4. The QBC algorithm is provided in Appendix B.4. Check them for details.
>
> (2) End-to-end RL
>
> We have trained the query proposer on Bunel et.al [1] jointly with a synthesizer at the very beginning which gets 40% on exact match, comparable to the result of 41.12%. However, the biggest drawback of this paradigm is that the query proposer is strongly correlated with the synthesizer, and the performance of the query proposer cannot be guaranteed once a new kind of synthesizer is proposed. Thus, we propose F-space and utilize the mutual information to encourage the query proposer to select queries based on the functionality of programs extracted from input-output examples instead of the feedback from a specific synthesizer (that is, decouple the query proposer from the synthesizer).
>
> Even though, the referential game is an interesting task that has a subtle but reasonable connection with the query problem. As we understand, the query network corresponds to the speaker which can observe what has happened in an oracle (i.e. program) and tries to send information (input-output examples) to the listener to finish the task (program synthesis). And the synthesis network corresponds to the listener, which receives the information sent from the query network and finishes the task based on it. We may try to apply the F-space to the referential game inversely in the future.

---

> > ### Author Response · Authors · 2021-11-18
> > **Response to Reviewer WAY8 (2/2)**
> >
> > ### Related work
> >
> > We have supplemented these two works in our related works. Considering that these two works are a series of works to find a hypothesis (program) by selecting informative observations (input-output examples), we add them into a new paragraph "learning to acquire information". See Section 5 for details.
> >
> > ### Questions
> >
> > (1) As you have seen, for the sampling of queries, we use Gumbel softmax to make this process differentiable. For the output of the oracle, we take it as a kind of RL problem, and thus the output can be seen as the “next state” (input-output examples) received from the “environment” (program). However, different from the RL problem, our supervision is not obtained by assigning rewards (which is hard to design it suitably) but calculated from the mutual information.
> >
> > (2) We obtain negative samples the same as the SimCLR [2] does: in a single training batch with N training samples, only the corresponding ([[e]], p) are taken as positive pairs, and the rest are taken as negative pairs. That is, we have N positive pairs and N*(N-1) negative pairs in one batch.
> >
> > (3) Your suggestion is quite reasonable and may be the future work of F-space. In this work, we focus on how to tackle the query problem without much influence on the original program synthesis techniques, so we decoupled the training of query network and program synthesis network, and present a way to encode the functionality of the program to F-space with distribution embedding and contrastive learning. However, as you said, the decoding process of F-space is also an interesting problem. Though, as we understand it, this process may suffer the problem of DSL: different DSLs have different representation abilities but they represent the same F-space, so how to decode the program into a suitable DSL? A feasible way to do this is to give a naïve DSL at the beginning of decoding, and let the DSL grow up to a suitable one automatically using library learning [3, 4]. This is our rudimentary thinking and we are really interested in finishing this work to show the advantage of F-space in the future.
> >
> > (4) We have rephrased “actual distribution” to “training distribution” for a better understanding.
> >
> > [1] Leveraging Grammar and Reinforcement Learning for Neural Program Synthesis
> >
> > [2] A Simple Framework for Contrastive Learning of Visual Representations
> >
> > [3] Library Learning for Neurally-Guided Bayesian Program Induction
> >
> > [4] DreamCoder: Growing generalizable, interpretable knowledge with wake-sleep Bayesian program learning

---

> > ### Comment · Reviewer_WAY8 · 2021-11-22
> > **thanks.**
> >
> > I asked for a baseline of QBC, I have received it. I will increase my score from 6 to 7. I do not mind that the performance of QBC-crash-aware (a very simple baseline that you implemented in a week, as opposed to however long it took you to do your original approach) is dangerously good when compared against your approach, I respect the honesty of the authors showing us the results. thank you.
> >
> > The question remain is how would you improve the current approach so that it may perform significantly better than QBC? I feel the answer lies in the negative sampling step of question (2), which I do not think you fully answered. Given a set of examples, we want to sample negative programs that are not the target program, but nonetheless consistent with the examples so-far. Failure to do this will cause the query network to not be maximally informative _conditioned on_ the examples given so-far. I'd be interested to see how you will manage to do that, because sampling negative programs consistent with a set of examples is a very difficult problem, yet one that must be solved for your approach to truly make sense.
> >
> > w.r.t. this work, I feel a good "demo" figure of sorts showing a concrete example (like figure13) would be a good idea. Imagine figure1 but make it very concrete, under the context of Karel the robot. Most other reviewers missed what I think is the central premise of this work, which is a method to ensure the train/test distribution in a synthesis system to be same, and yet still in a way not alien to end-users, and instead went off nitpicking the implementation of the F-space, which, since it is all neural anyways, it doesn't really matter.
> >
> > Finally, I do invite the authors to build an analogy of synthesis and communication, as there are many under explored problems in that space. Specifically, one can think of communicating under consistency of random input-output as communication over literal semantics (sat / unsat), and communication under consistency _and_ choice of input-output as pragmatics (meanings + context), and the end-to-end RL communication to be consistency+pragmatics+convention, i.e. the speaker and listener collude upon a shared convention knowable to the pair alone, and unintelligible to any other agents. The following is a cute paper that you might find interesting https://arxiv.org/pdf/2107.00077.pdf .
> >
> > best of lucks in your research endeavors, this area is very good.

---

> > > ### Author Response · Authors · 2021-11-23
> > > **The response to two questions.**
> > >
> > > Many thanks for your suggestion on the negative sample selection, writing, and future research! However, there are still two questions that we need to clarify.
> > >
> > > (1) The performance of QBC.
> > >
> > > Actually, we do not think that QBC-crash-aware performs as well as our method. As we stated in "Explanation" and the revised paper, to get this performance, QBC-crash-aware filters all the crashed queries out. That means, QBC-crash-aware queries much more times for "whether this query will crash when executing" than our method. For example, suppose there is a user who wants to synthesize a program, our method simply asks him/her for 5 queries and then synthesizes the underlying program. For QBC-crash-aware, however, it asks: "What is the response of this query?", the user: "It is crashed", "What about this one?", "Crashed", ......, "This one?" "A valid query finally", which is more resource-consuming. To measure this quantitatively, we list the crashed times before a successful query of QBC-crash-aware as follows:
> > >
> > > | query step | 1    | 2    | 3    | 4    | 5    |
> > > | ---------- | ---- | ---- | ---- | ---- | ---- |
> > > | avg        | 5.72 | 2.47 | 2.96 | 4.27 | 4.86 |
> > > | max        | 214  | 200  | 224  | 357  | 226  |
> > > | min        | 0    | 0    | 0    | 0    | 0    |
> > >
> > > In conclusion, QBC-crash-aware performs close to our method only when it can get much more privileged information, and thus we think that our method performs significantly better than it (after all, we even outperforms the well-designed input-output examples with high branch coverage).
> > >
> > > Looking forward to more discussions here if you have a different idea.
> > >
> > > (2) The sampling approach.
> > >
> > > Though we think the performance of QBC is a misunderstanding, we should still make a discussion on our sampling method. Actually, "Failure to do this will cause the query network to not be maximally informative *conditioned on* the examples given so-far" is not correct, and we will illustrate this as follows:
> > >
> > > Specifically, what we need to optimize is the mutual information conditions of the next query conditioned on the query history I(p;q_{t+1}|[[q]]_t) (as you said). What is really optimized here is the mutual information between all queries up to now and the program I(p;[[q]]\_t+q\_{t+1}), which seems inaccurate. However, note that we apply an "RNN-style" training process, where we optimize the InfoNCE loss between queries and the program at every query step, see Figure 3. This training process guarantees that what we optimized is the information gain of each query step instead of a total information gain of a set of queries, and thus the conditioned mutual information is optimized during training.
> > >
> > > As we understand it, the sampling problem you mentioned is closer to "the problem of the quality of dataset" instead of a wrong optimization objective. For example, in image encoding, the representation of the image learned by contrasting demi-wolves with huskies is much better than the representation learned by contrasting dogs with cats. So your advice that we should sample negative samples more elaborately definitely can improve the performance of our method, but the only problem is that such a dataset is really hard to generate. To bypass this problem, we have a new idea which is under-explored, that we can abandon the popular InfoNCE and estimate a new lower bound of the mutual information, with which we do not need to construct negative pairs. More details remain to be checked.
> > >
> > > Finally, thanks again for your advice, and we will revise our paper as you suggest!

---

> > > > ### Comment · Reviewer_WAY8 · 2021-11-24
> > > > **let's see . . .**
> > > >
> > > > ## on 1
> > > >
> > > > I think the most solid move here is to treat crash just as any other form of information and do synthesis over crash anyways. Imagine I come up to a synthesizer and says, please come up with a function that:
> > > >
> > > > - 8 -> 3
> > > > - 2 -> 1
> > > > - 1 -> 0
> > > > - 1/2 -> -1
> > > > - -3 -> crash
> > > >
> > > > then maybe a reasonable answer would be, that is the log(x) function, which you clearly hinted to me that it is undefined over negative numbers. so really, there is mutual information between the crash outcome and what the program is, and there is no problem treating it like any information as you would. in fact, the question "does it crash on negative numbers?" is very informative first question one could ask no?
> > > >
> > > > on the implementation side, for the very very short-term, you can do something like... I'm assuming your synthesizer is of the form "generate and check" which consisting of a generative model that, conditioned on some spec, propose a number of programs, you then explicitly execute each proposed program to see if they match the input-output pairs in the specification. you can perhaps ignore the crash information for generation (i.e. don't condition your NN program generator on the crash information -- this way you don't have to re-train anything) but during the checking step make sure the proposed program correctly crash on the same query that crashed the oracle program.
> > > >
> > > > however, a more solid approach would really be to re-train everything with crash as a special value, maybe your query network would learn to query for crashes, which would be pretty interesting in its own right. I've always had a hunch the "valid domain" for a function would pop up somewhere as being important thing to be communicated, as almost no computer programs are total, so that'll be kinda cool.
> > > >
> > > > ## on 2
> > > > let me read the supervised contrastive learning paper again to get a better sense and get back to you. I think I understand what you are saying and I tentatively agree that, it is the right objective if you do it for long enough, but it's just the quality of samples will degrade (exponentially so as each query prunes a fixed fraction of valid programs) rather quickly

---

> > > > > ### Author Response · Authors · 2021-11-25
> > > > > **The short-term implementation**
> > > > >
> > > > > Thanks for your comments, and we have conducted a new experiment on QBC as you suggested.
> > > > >
> > > > > In your short-term method, the first query should not crash, because if so the program synthesis network cannot receive it as valid input. And, it is a bit hard to get the first query (we have generated for hours with multi-processes to find a query that does not crash on all training programs as the first query but failed), so we still keep the first query with higher quality by resampling it for each program until no crash happens. Here is the result.
> > > > >
> > > > > | query step             | 1      | 2      | 3      | 4      | 5      |
> > > > > | ---------------------- | ------ | ------ | ------ | ------ | ------ |
> > > > > | exact match            | 29.31% | 33.31% | 34.82% | 35.58% | 36.54% |
> > > > > | functional equivalence | 34.03% | 39.10% | 41.89% | 43.21% | 44.05% |
> > > > >
> > > > > As you suggest, in the future we will retrain both the synthesis network and the query network to implement a query method that can query for crashes. Actually, for the query network, this may be easy because our query method does not care about the execution status of programs, but for the synthesizer, no one has ever done this before and some synthesis networks are guided by the program execution, so we cannot guarantee the program synthesis performance after retraining.

---

### Official Review · Reviewer_Ytee · 2021-11-03

**Correctness:** 3
**Technical Novelty And Significance:** 3
**Empirical Novelty And Significance:** Not applicable
**Recommendation:** 3
**Confidence:** 4

**Main Review:**

Strengths:
- The interactive framing of the problem is useful to explore. For program synthesis from input-output examples, there often is an implicit oracle for the unknown program that the user can use, even if it is not cheap to compute. If the user has to decide the input-output examples without guidance, then the set may not end up being very informative at distinguishing between possible programs. By having the synthesizer query the user for inputs interactively, it can obtain the specification most useful for generating the right program.
- The F-space formalism is a nice way of thinking about the space of programs and functional equivalence, although it is not practical (as it requires enumerating all possible inputs to the programs) and the definition seems unlikely to be novel in the literature.

Weaknesses:
- The modeling of F-space and sets of input-output examples as normal distributions seemed inappropriate. In particular, equation 3 states that we want to model the intersection of two sets by intersecting the distributions (taking a product of the pdfs?) but then equation 4 goes onto take a weighted sum of the mean and log of the variance parameters. Equations 4 and 5 cite Ren & Leskovec 2020, but that paper took weighted sums of Beta distribution parameters, not normal distribution parameters.
- The paper does not provide all details necessary to reproduce the results (see suggestions for more details).
- The paper doesn't describe what happens for invalid inputs to programs (i.e. inputs which cause the programs to crash). For example, in Karel, it is not allowed to take a marker from a cell containing no markers.

Suggestions:
- Provide greater details necessarily for reproducibility of the paper. Various details that were necessary to obtain the results are obscure in the appendix and not described elsewhere. Examples:
  - Query decoder for Karel: if it is similar to the query encoder, how does it generate a grid world state as the output?
  - Query decoder for list processing: since input examples
  - "additional batch normalization is added to keep the generation more stable": how was this done?
  - "we introduce a latent code like InfoGAN": this can be described more precisely, including with equations
  - "we add the reciprocal of the KL divergence [...] this loss does not have a significant impact on training most of the time and makes the
training process unstable": If the loss was not useful, then was it still kept?
  - "At the beginning of the training, the query network generates one query only. As the training goes on, the number of query times increases until it achieves the query time limit.": How was the number of generated queries increased over time? Why is the curriculum necessary?
  - "to guarantee that every query can be recognized by the program simulator, we design the output layer of the query network elaborately.": it is not clear what was done in the output layer to guarantee this.
- Add a prior to F-space and use it to generate the best queries. The set of all possible programs is very large or infinite, but the set of useful programs that the user might want is smaller. It should be possible to learn the prior distribution (or assume one like Occam's razor) and use it to inform the next query.

**Summary Of The Paper:**

In neural program synthesis from input-output examples, the goal is to generate a program which, when executed on the inputs, produces the corresponding outouts. Typically, the set of input-output examples is taken as given by the program synthesis method. In this work, the authors propose to instead have the program synthesizer interact with an oracle to determine the set of input-output examples. The synthesizer proposes a potential input for the program, the oracle produces the output, and the process repeats. Under this paradigm, the program synthesizer can query the oracle for the inputs which would be most helpful in determining the correct program. This direction has been explored in prior work as "interactive program synthesis", but largely with constraint-based methods.

The paper represents each input-output example pair as a normal distribution (with a mean and variance parameter). A set of input-output example pairs is also represented as a normal distribution, with its parameters a weighted sum of each pair's parameters. A neural decoder module takes the mean/variance of this combined distribution as input to produce a new input; we can query the oracle with the input to get the output, resulting in a new input-output example pair.


**Summary Of The Review:**

The paper has a useful motivation and explores interesting idea, but the aspects of the execution and the missing details in the work make it difficult to recommend acceptance.

---

> ### Author Response · Authors · 2021-11-18
> **Response to Reviewer Ytee (1/2)**
>
> We thank the reviewer for the thorough comments. We have supplemented all details you mentioned. However, many details cannot be explained here easily, so please check our revised paper.
>
> Q1: The modeling of F-space and sets of input-output examples as normal distributions seemed inappropriate. In particular, equation 3 states that we want to model the intersection of two sets by intersecting the distributions (taking a product of the pdfs?) but then equation 4 goes onto take a weighted sum of the mean and log of the variance parameters. Equations 4 and 5 cite Ren & Leskovec 2020, but that paper took weighted sums of Beta distribution parameters, not normal distribution parameters.
>
> A1: According to [1], any distribution in an exponential family satisfies the requirement of using the product of distribution to approximate the intersection of sets, and the Normal distribution is one of them. To make this clearer, we have supplemented more details in Appendix C, including the proof that the product of two Normal distributions is a scaled Normal distribution, and expanding the mu and log(sigma^2) of the new Normal distribution to show that they can be approximated by weighted sum of the two original distributions the same as the Beta distribution.
>
> Q2: The paper does not provide all details necessary to reproduce the results
>
> (1) Query decoder for Karel
>
> A: We have added a figure to illustrate the details of the query network’s architecture in Karel. See Appendix A.2 and Figure 6.
>
> (2) Query decoder for list processing
>
> A: We have added a figure to illustrate the details of the query network’s architecture in list processing. See Appendix A.2 and Figure 7.
>
> (3) "additional batch normalization is added to keep the generation more stable": how was this done?
>
> A: As mentioned above, this can be seen in Appendix A.2 and Figure 6.
>
> (4) "we introduce a latent code like InfoGAN"
>
> A: We have added more descriptions including equations in this paragraph to make this statement clearer. How is the latent code added to the overall architecture can be seen in Figure 6 and Figure 7.
>
> (5) "we add the reciprocal of the KL divergence [...] this loss does not have a significant impact on training most of the time and makes the training process unstable": If the loss was not useful, then was it still kept?
>
> A: Sorry for the misunderstanding, this loss is not used. Here we just want to present the results of several techniques to help future improvements. We have added “Thus, this loss is not used in our final version” to the end of this paragraph to make it clear.
>
> (6) "At the beginning of the training, the query network generates one query only. As the training goes on, the number of query times increases until it achieves the query time limit.": How was the number of generated queries increased over time? Why is the curriculum necessary?
>
> A: We have added more details to this paragraph. We conduct the curriculum learning by increasing the steps of querying from 1 to 5 by 1 every 2 epochs. Curriculum learning is necessary because it helps the network learn short queries better before learning long queries, which is much easier.
>
> (7) "to guarantee that every query can be recognized by the program simulator, we design the output layer of the query network elaborately.": it is not clear what was done in the output layer to guarantee this.
>
> A: We have added more details to illustrate this in the architecture design of Appendix A.2. In short, we split the generated query into several parts and each part is easy to be generated without illegal actions. Then, we concatenate each part to generate the final query. Specifically, in Karel, we split the query into "boundaries", "hero position", and "map situation", and generate each of them separately. In list processing, we split the query into "INT", "LIST", and "NULL", and then select among them based on the input types.

---

> > ### Author Response · Authors · 2021-11-18
> > **Response to Reviewer Ytee (2/2)**
> >
> > Q3: The paper doesn't describe what happens for invalid inputs to programs (i.e. inputs which cause the programs to crash). For example, in Karel, it is not allowed to take a marker from a cell containing no markers.
> >
> > A3: We have added these details in Appendix B.1 “handling of the crash problem” and B.2 “handling of the out of range problem”. In conclusion, we modify the executor to ignore crashes when training the query network. As the training goes on, the crashes reduce since crashed queries cannot provide any useful information for the training of the query network.
> >
> > Q4: Add a prior to F-space and use it to generate the best queries. The set of all possible programs is very large or infinite, but the set of useful programs that the user might want is smaller. It should be possible to learn the prior distribution (or assume one like Occam's razor) and use it to inform the next query.
> >
> > A4: This prior can be learned by training the query network with the dataset that consists of meaningful programs. That is, the programs in the dataset form a program distribution, and conducting training on this specific program distribution can help the network to learn this distribution as its prior. In this paper, we just take the program distribution in the original dataset as our prior. If a program looks strange and out of this distribution, then it is less likely to be synthesized.
> >
> > [1] Information-Geometric Set Embeddings (IGSE): From Sets to Probability Distributions

---

> ### Comment · Reviewer_WAY8 · 2021-11-22
> **I agree with this**
>
> "The F-space formalism is a nice way of thinking about the space of programs and functional equivalence, although it is not practical (as it requires enumerating all possible inputs to the programs) and the definition seems unlikely to be novel in the literature."
>
> I definitely think F-space is just version space, and would probably be best explained as such. That being said, such space probably _could_ be modeled and will be practical if you train a large enough NN, especially under the infinite-data regime of synthesis, where given a DSL you can generate / train programs for as long as you want (in the limit, you would have memorized the whole program space). So given recent advances such as codex showing it can somewhat learn a complicated space like python, for Karel I strongly believe if you just train a big enough NN it might be able to flush out the entire version space, i.e. {programs consistent with any set of examples} and be able to perform set operations on these version spaces as we have in symbolic approaches.
>
> curious to see what you think of it though aha

---

> > ### Author Response · Authors · 2021-11-23
> > **Thank you for your comments, here's our response to this question.**
> >
> > Thank you for your comments! We reply to these two questions as follows.
> >
> > Q1: "the definition seems unlikely to be novel in the literature" and "I definitely think F-space is just version space".
> >
> > A: If what you mean by saying version space is the machine learning concept then yes, in the sense of representing the programs that satisfy given input-output examples, F-space can be considered as a version space. However, we think that the version space mentioned here is just a high-level concept more than a practical tool that can be used directly. And if the version space you said implicates the version space algebra in the program synthesis tools like SMARTedit [1], we think that there are several differences between F-space and it, which we summarized as follows:
> >
> > (1) Distance. In F-space, we can define distance/similarity that is related to the program semantics while in version space algebra the definition is not simple. In our paper, we define the distance between two different programs as the number of different input-output examples and the similarity between a set of input-output examples and a program as the probability in a parameterized distribution that is differentiable. Under this definition, we can model the mutual information and train the network to optimize it, which we think is surely novel and the version space algebra definitely cannot do it.
> >
> > (2) DSL. The construction of the version space relies on an initial DSL, which can represent the whole "functionality space". However, as for F-space, we design it to have the ability to handle the situation that is DSL-agnostic by using input-output examples to represent a functionality due to the fact that the same F-space can be expressed by different DSLs, and these different DSLs can lead to very different results in downstream tasks [2] which is unacceptable. To do this, we can conduct the contrastive learning between input-output examples using KL divergence without explicit programs written in a DSL, and the information of the functionality of the programs is brought to the neural network by executing input examples.
> >
> > Q2: "it is not practical"
> >
> > A: This can be answered by Q4 & A4. Typically, you are right that a larger dataset and a bigger model may get a better result. Even though, we have a different perspective that, we assume that the distribution of the programs in the training set keeps the same as the one during testing, and what we trained is what we need. Specifically, an ideal F-space should be obtained by traversing all inputs for all programs which is impossible as you said. However, most of the programs are not commonly used in reality, and we take the programs that appeared in the training set as the programs that are frequently used (i.e. we assume that the program distribution in training is similar to the one in testing). After training, the learned "F-space" can identify programs that are drawn from the training distribution perfectly, but it may not have the ability to tell the difference among programs with low probability in the training distribution. For example, the difference between "a descending order sorting algorithm" and "an ascending order sorting algorithm" is obvious and should be modeled by F-space. But the difference between "a descending order sorting algorithm" and "a descending order sorting algorithm with the last element changed to 0 if this program is running on a V100 GPU on Sunday evening ......" should not be modeled because the model capacity is limited and almost no user needs the later program. The learned F-space just takes the later as "a descending order algorithm".
> >
> > Looking forward to more discussions!
> >
> > [1] Version Space Algebra and its Application to Programming by Demonstration
> >
> > [2] Benchmarking Meaning Representations in Neural Semantic Parsing

---

### Official Review · Reviewer_KXs9 · 2021-11-04

**Correctness:** 2
**Technical Novelty And Significance:** 2
**Empirical Novelty And Significance:** 1
**Recommendation:** 3
**Confidence:** 5

**Main Review:**

The problem of generating appropriate unit tests in an interactive manner and has the potential to be a key ingredient in future programming workflows. However I think the paper as written needs improvement before it can be accepted. In particular, I think the following points need attention :-

(1) They say that the Product of normal distributions is a normal distribution but that's not necessarily the case in general - for example see https://math.stackexchange.com/questions/101062/is-the-product-of-two-gaussian-random-variables-also-a-gaussian/101120#101120

(2) Sanity checks :-
  (a) I would expect the merged unit test distribution to be lower entropy than the distribution of the individual unit tests (as more unit tests mean the functions satisfying them are fewer). This is something that should be examined.
  (b) They don't test whether the program they synthesize in the end has a high probability under the merged unit test normal distribution vs a  program that doesn't satisfy those unit tests

(3) Interactive unit test generation has been looked at in the past and it's unclear the different ways in which their setup differs. It's also unclear what assumptions they are making that guide their design choices (along with theoretical or empirical validation for those assumptions). Some illustrating examples would help develop their intuition here.

(4) They should compare against more baseline methods. For example they mention Padhi et al. which present multiple queries to the user and let the user pick which one to label. This they claim puts additional burden on the user. But they could still compare against Padhi et al. in a way that's fair to them by selecting for example a query at random for the user to label out of the ones that Padhi et al. suggest.

(5) The empirical results with the baselines they compare against aren't very compelling. For example they are much worse in the Searching for Semantics metric for list processing task compared to the baseline methods and only very marginally better in the searching for exact match metric. They also don't give error bars for their results.

**Summary Of The Paper:**

The authors propose a framework to generate unit tests by generating candidate inputs for which labels are queried from an oracle. The idea is to have a vector space where each point represents a function (different programs that are functionally equivalent map to the same point) and a unit test is represented as a set of points that correspond to functions that would pass that unit test.

They project a unit test onto a normal distribution on the function space where the probability of the distribution is whether a point is the underlying function to be synthesized given the unit test.

They merge multiple unit tests using an attention mechanism into a single point in the function space and then train the unit test generator such that the mutual information between the probability distribution on the merged unit test and the probability distribution on the programs is maximized using InfoNCE loss.

A synthesizer network is used to synthesize the program from generated unit tests. They test on Karel and list processing task.

**Summary Of The Review:**

The problem is interesting but the paper suffers from some incorrect claims, insufficient motivation and justification, and underwhelming empirical results.

---

> ### Author Response · Authors · 2021-11-18
> **Response to Reviewer KXs9 (1/3)**
>
> We thank the reviewer for the comments and clarifying questions.
>
> We notice that you take our task as a unit test generation problem, which is a misunderstanding. Before responding to your questions, we should clarify that our work should not be associated with the “unit test generation” task at all. The unit test aims to verify the functionality of a piece of a program using several input-output pairs and it is more relevant with the “functional equivalence” as an evaluation criterion, while our goal is to find the underlying program inside the oracle by querying it.
>
> Specifically, we can classify the unit test into two categories: white-box testing and black-box testing. For the white-box testing, the program under test is given which is obviously different from our "underlying program" setting where the program acts like an oracle that cannot be seen. For the black-box testing, although the program acts like a black-box whose internal structures cannot be peered into, the settings and the purpose of this task is totally different from ours. In detail, black-box testing assumes the existence of a target requirement (or target function) and a black-box implementation of this requirement (such as a piece of program which is unknown). The purpose is to check the black-box implementation to ensure that there is no bug or difference between the target requirement and the black-box by generating test cases (say input-output examples). As a contrast, in our settings, the target function does not exist, and what we have is only a black-box (or called oracle / underlying program in our paper). Our goal is to query this black-box and guess which program is inside based on the experience learned on the training set.
>
> To be more specific, here is an example. Suppose the target requirement is "a sorting algorithm". For the black-box testing, this sorting algorithm is already known, and there is a black-box that implements it in without showing its structure. There may be bugs or mishandlings in this black-box, and the black-box testing needs to generate some test cases to diagnose the black-box and make sure that it works well. If the black-box fails to pass the test cases, then it is thought to be a bad implementation. However, for the query problem in our paper, we do not know the target requirement, and what we have is just a black-box. We don't have any information about the black-box, and what we do is try to guess what the black-box is by querying. After some queries, we guess that "it is a sorting algorithm" based on the information brought by querying and our learned experience. If the black-box is not a sorting algorithm, it does not mean that the black-box is implemented badly, it means that what we guessed is wrong.
>
> In a word, the unit test aims to judge the program’s quality by generating input-output examples while the query problem aims to identify an unknown program actively based on the learned experience.
>
> We will show an experiment example on branch coverage (the metric of the unit test) to illustrate this further in Q3, and for a better understanding of the query problem, we have added a problem statement in Section 2.1 and a detailed comparison between the black-box testing and the query problem in Appendix D.2.

---

> > ### Author Response · Authors · 2021-11-18
> > **Response to Reviewer KXs9 (2/3)**
> >
> > Q1: They say that the Product of normal distributions is a normal distribution but that's not necessarily the case in general
> >
> > A1: This link has nothing to do with the product in our work at all. The question in this link shows that the distribution of the product of normal variables may not be a normal distribution (i.e. p(x*y) is not a Normal distribution in some cases). However, what we do in our work is get a new distribution by multiplying several normal distributions, which results in a new normal distribution (i.e. p_a(x)*p_b(x) is a normal distribution). Equation (3) in the paper clearly shows that what we calculated is the product of Pr, not the Pr of product.
> >
> > A more detailed proof is given in Appendix C and more details of why this product is reasonable can be seen in [1, 2].
> >
> > Q2: Sanity checks :- (a) I would expect the merged unit test distribution to be lower entropy than the distribution of the individual unit tests (as more unit tests mean the functions satisfying them are fewer). This is something that should be examined. (b) They don't test whether the program they synthesize in the end has a high probability under the merged unit test normal distribution vs a program that doesn't satisfy those unit tests
> >
> > A2: (a) We have added this experiment in Appendix B.3. The entropy of a multivariate Normal distribution is correlated with the determinant of its covariance matrix and its dimension. The dimension of our Normal distribution is fixed, and under the assumption of independence of each dimension, we calculate the sum of log(sigma^2) as the determinant of the covariance matrix. The result shows that the entropy decreases generally as the query goes on. (b) This is surely satisfied because of the loss presented in Equation (7). That is, if the loss is small, then it is more likely that the right program has a high probability and the other programs have a low probability.
> >
> > Q3: Interactive unit test generation has been looked at in the past and it's unclear the different ways in which their setup differs. It's also unclear what assumptions they are making that guide their design choices (along with theoretical or empirical validation for those assumptions). Some illustrating examples would help develop their intuition here.
> >
> > A3: As stated at the beginning, the query problem is totally different from the unit test generation problem. For the guidance of our query choices, according to Equation (7), we choose our query based on the mutual information between our queries and the underlying program. To achieve this, we train our query network by letting it maximize the mutual information under the formulation of F-space which models the relationship between queries and programs. We have modified Section 2 to make the problem statement clearer.
> >
> > An experimental example: Consider the Karel task, the dataset obtained by our query method has only 72.27% branch coverage while the well-designed dataset has 87.99% coverage. However, as we presented in our paper, the queried dataset achieves comparable results with the well-designed dataset and is even better. Thus, the branch coverage is not the only factor in determining the performance of the query, while in unit test generation, the input-output examples with high branch coverage is their final purpose.

---

> > > ### Author Response · Authors · 2021-11-18
> > > **Response to Reviewer KXs9 (3/3)**
> > >
> > > Q4: They should compare against more baseline methods. For example, they mention Padhi et al. which present multiple queries to the user and let the user pick which one to label. This they claim puts additional burden on the user. But they could still compare against Padhi et al. in a way that's fair to them by selecting for example a query at random for the user to label out of the ones that Padhi et al. suggest.
> > >
> > > A4: We compare our method with another baseline: query by committee (QBC) [3] mentioned by reviewer WAY8, which is much suitable for our experiments for its scalability on different tasks. The results are shown as follows.
> > >
> > > Exact match:
> > >
> > > | Query step        | 1      | 2      | 3      | 4      | 5      |
> > > | ----------------- | ------ | ------ | ------ | ------ | ------ |
> > > | Query (ours)             | 23.52% | 37.48% | 38.08% | 38.16% | 41.12% |
> > > | Random            | 15.80% | 22.40% | 26.52% | 28.40% | 30.12% |
> > > | QBC-crash-aware   | 29.75% | 34.19% | 36.06% | 36.38% | 36.34% |
> > > | QBC-crash-unaware | 11.43% | 12.86% | 11.98% | 7.90%  | 6.55%  |
> > >
> > > Functional equivalence:
> > >
> > > | Query step        | 1      | 2      | 3      | 4      | 5      |
> > > | ----------------- | ------ | ------ | ------ | ------ | ------ |
> > > | Query (ours)            | 27.16% | 42.28% | 43.04% | 43.08% | 46.64% |
> > > | Random            | 18.16% | 25.16% | 29.52% | 31.68% | 34.00% |
> > > | QBC-crash-aware   | 34.46% | 40.54% | 43.85% | 44.97% | 44.97% |
> > > | QBC-crash-unaware | 13.33% | 20.48% | 23.61% | 21.35% | 21.07% |
> > >
> > > Explanation:
> > >
> > > We trained a program synthesizer the same as Bunel et. al[4] except that it can receive 1-5 IOs instead of 5 IOs only. The query by committee algorithm executes as follows: given {(x_1, y_1), …, (x_{t-1}, y_{t-1})}, we use the synthesizer to generate top-100 programs by beam search. Then, we select 100 inputs from the training set randomly (just take it as an input generator that will not suffer the out-of-distribution problem) and feed them into 100 programs to get 100x100 outputs. We select the input that has the most diversity (the more different outputs given on the 100 programs, the higher the diversity) as the next query x_t. The first query is selected randomly from the training set without scoring.
> > >
> > > In program synthesis, illegal queries will cause the crash (or out-of-range) problem that the program may terminate and return an error (See Appendix B.1 and B.2). So there are two strategies based on QBC: Crash-aware, where if the query crashes then repeat the algorithm to sample another one; And crash-unaware, which samples queries regardless of the crash problem. QBC-crash-unaware performs worse than Random because Random is chosen by filtering crashed IOs while QBC-crash-unaware may contain crashes. QBC-crash-aware performs much better than QBC-crash-unaware because it queries the underlying program multiple times to make sure that the query will not result in a crash, which is unfair. Even though, our method still outperforms QBC-crash-aware, which shows its advantage. We list the crashed times before a successful query of QBC-crash-aware as follows to show our adantage more clearly:
> > >
> > > | query step | 1    | 2    | 3    | 4    | 5    |
> > > | ---------- | ---- | ---- | ---- | ---- | ---- |
> > > | avg        | 5.72 | 2.47 | 2.96 | 4.27 | 4.86 |
> > > | max        | 214  | 200  | 224  | 357  | 226  |
> > > | min        | 0    | 0    | 0    | 0    | 0    |
> > >
> > > We have supplemented all results to Experiment 4.2 " Comparison with others" and Figure 4. The QBC algorithm is provided in Appendix B.4. Check them for details.
> > >
> > > Q5: The empirical results with the baselines they compare against aren't very compelling. For example they are much worse in the Searching for Semantics metric for list processing task compared to the baseline methods and only very marginally better in the searching for exact match metric. They also don't give error bars for their results.
> > >
> > > A5: As shown in the introduction, the query for program synthesis is much harder but more practical than using the well-designed input-output examples. Even so, we outperform the well-designed method in Karel which has a large input-output space. The results in list processing are barely satisfactory because instead of seq2seq methods in Karel, the principle of the list processing task is to synthesize programs by a neural guided search, which has a larger gap with our neural method. In Appendix A, we can see that the architecture of the list processing task is much simpler than the one in Karel because how to use an end-to-end seq2seq method to finish the list processing task is unexplored.
> > >
> > > [1] Information-Geometric Set Embeddings (IGSE): From Sets to Probability Distributions
> > >
> > > [2] Sided and symmetrized Bregman centroids
> > >
> > > [3] Query by committee
> > >
> > > [4] Leveraging Grammar and Reinforcement Learning for Neural Program Synthesis

---

### Author Response · Authors · 2021-11-18
**Paper Revision**

Dear reviewers, we would like to thank you for all your comments and suggestions. We have updated our paper with a revision to address them. We summarize the main changes as the following:

(1) [Abstract] “actual distribution” -> “training distribution”.

(2) [Overview] We add a “problem statement” section before the “problem formulation” and change the title of Section 2 from “problem formulation” to “Overview”.

(3) [Related Work] Two more related works added to a new paragraph: Learning to acquire information. The original subsection "Active Learning" is moved to Appendix D.1 due to the page limit.

(4) [Experiment 4.2] The comparison with a new baseline method: query by committee, is added. The result is shown in Figure 4 (two lines are added).

(5) [Algorithm 1] Algorithm 1 is moved to Appendix A.1 due to the page limit.

(6) [Appendix A.2 A.3] Sufficient details of the query networks, including architectures and training techniques (latent code, curriculum learning, and KL divergence).

(7) [Appendix B.1] The handling of crashes is added.

(8) [Appendix B.2] The handling of the out-of-range problem is added.

(9) [Appendix B.3] The experiment on the change of the distribution entropy during query is added.

(10) [Appendix B.4] The algorithm details of our new baseline: query by committee (QBC).

(11) [Appendix C] Additional proof on the product of two Normal distributions and that it can be approximated by the weighted sum of the original mu and log(sigma^2).

(12) [Appendix D] Related work: "Active Learning" is moved to Appendix D due to the page limit. "Automated Black-box Testing" is added to make the statement of the query problem clearer.

(13) [Appendix E] Remove two Karel examples from Appendix B.2 to Appendix E.

---

### Decision · Program_Chairs · 2022-01-20

**Decision:**

Accept (Poster)

**Comment:**

The paper addresses an important problem of selecting inputs to drive an inductive program synthesis process. This is an important problem because inductive synthesis relies on carefully chosen inputs to ensure that the chosen inputs can provide sufficient information about what the desired program is. This paper proposes an approach where instead of simply asking the user to provide a set of input/output examples to the synthesizer, the user interacts with a query network that queries the user on the output of specific inputs and these input/output examples are then fed into an existing program synthesis engine.

I think the ideas in the paper are very original and I agree with reviewer WAY8 that this paper should be accepted. I think the original version of this paper had several issues that led to the low scores from the other reviewers, but the paper improved significantly with the review process.

That said, I do think that some of the concerns of the other two reviewers are valid, and some additional steps could be taken to address them. For example, the paper follows a long tradition in ML of making assumptions that are questionable but that make the math work nicely (e.g. choosing to represent things as gaussian distributions, or assuming independence for things that are clearly not independent). We are usually ok with such shortcuts if they are properly acknowledged and the resulting method proves to work well empirically, but the original set of experiments in the paper was extremely minimal. That said, the experiments added through the rebuttal process give me more confidence that even with the mathematical shortcuts, the method still works well.